# Evolutionary divergence reveals the molecular basis of EMRE dependence of the human MCU

Melissa JS MacEwen[1],*, Andrew L Markhard[2],*, Mert Bozbeyoglu[2], Forrest Bradford[1], Olga Goldberger[2], Vamsi K Mootha[2,3,4], Yasemin Sancak[1]

The mitochondrial calcium uniporter (MCU) is a calcium-activated calcium channel critical for signaling and bioenergetics. MCU, the pore-forming subunit of the uniporter, contains two transmembrane domains and is found in all major eukaryotic taxa. In amoeba and fungi, MCU homologs are sufficient to form a functional calcium channel, whereas human MCU exhibits a strict requirement for the metazoan protein essential MCU regulator (EMRE) for conductance. Here, we exploit this evolutionary divergence to decipher the molecular basis of human MCU's dependence on EMRE. By systematically generating chimeric proteins that consist of EMRE-independent *Dictyostelium discoideum* MCU and *Homo sapiens* MCU (HsMCU), we converged on a stretch of 10 amino acids in *D. discoideum* MCU that can be transplanted to HsMCU to render it EMRE independent. We call this region in human MCU the EMRE dependence domain (EDD). Crosslinking experiments show that EMRE directly interacts with HsMCU at its transmembrane domains as well as the EDD. Our results suggest that EMRE stabilizes the EDD of MCU, permitting both channel opening and calcium conductance, consistent with recently published structures of MCU-EMRE.

## Introduction

Mitochondria play central roles in diverse cellular processes including metabolism, signaling, and cell death. Calcium ($Ca^{2+}$) signaling is critical for coordination of cellular needs with mitochondrial outputs by regulating the activity of the tricarboxylic acid cycle, activity of mitochondrial metabolite carriers, and triggering the mitochondrial permeability transition pore (Denton, 2009; Del Arco et al, 2016; Giorgio et al, 2018). This coordination is partially mediated by entry of $Ca^{2+}$ into the mitochondrial matrix from the cytosol during a $Ca^{2+}$ signaling event. Perturbation of mitochondrial $Ca^{2+}$ uptake is associated with a plethora of cellular and systemic pathologies, ranging from abnormal mitochondrial movement and shape, to immune dysfunction, cell

cycle progression, and neuromuscular disease (Logan et al, 2014; Mammucari et al, 2015; Prudent et al, 2016; Mammucari et al, 2018; Paupe & Prudent, 2018; Koval et al, 2019; Zhao et al, 2019).

The mitochondrial $Ca^{2+}$ uniporter complex, a multi-subunit protein assembly that resides in the inner mitochondrial membrane (IMM) is responsible for bulk entry of $Ca^{2+}$ ions into the mitochondrial matrix (Deluca & Engstrom, 1961; Vasington & Murphy, 1962; Carafoli & Lehninger, 1971; Kirichok et al, 2004). An ~35-kD protein termed mitochondrial calcium uniporter (MCU) is the defining component of the uniporter complex and serves as its pore (Baughman et al, 2011; De Stefani et al, 2011; Chaudhuri et al, 2013; Kovacs-Bogdan et al, 2014). MCU is a transmembrane protein with two membrane-spanning helices (TM1 and TM2), a short linker region facing the intermembrane space (IMS) with a highly conserved "DIME" motif, a large amino terminal domain that assumes a $\beta$-grasp fold (Lee et al, 2015, 2016), and a carboxyl terminal region that is mostly helical (Oxenoid et al, 2016; Baradaran et al, 2018; Fan et al, 2018; Nguyen et al, 2018; Yoo et al, 2018). Functional and structural studies have shown that TM2 forms the $Ca^{2+}$-conducting pore of the channel, whereas the N-terminal domain is mostly dispensable for $Ca^{2+}$ conductance and is likely to play a regulatory role (Lee et al, 2015; Oxenoid et al, 2016). In animals, MCU nucleates other proteins (MCUb, MICU1, MICU2, MICU3, and essential MCU regulator [EMRE]) that regulate different aspects of uniporter function. MICU1, MICU2, and MICU3 are EF-hand containing $Ca^{2+}$-binding proteins that localize to the IMS of the mitochondria. MICU homologs are not necessary for $Ca^{2+}$ conductance by the uniporter, but rather, they play crucial roles in setting the threshold for $Ca^{2+}$ uptake (Perocchi et al, 2010; Mallilankaraman et al, 2012; Csordas et al, 2013; Plovanich et al, 2013; de la Fuente et al, 2014; Foskett & Madesh, 2014; Patron et al, 2014; Liu et al, 2016; Kamer et al, 2018). In certain cell types, however, loss of MICU1 homologs leads to loss of other uniporter components, including MCU, leading to a decrease in uniporter activity (Plovanich et al, 2013). MCUb is a paralog of MCU and is thought to be a negative regulator of MCU because of its inability to form a functional Ca2+ channel (Raffaello et al, 2013).

EMRE is a single-pass transmembrane protein that was the last essential component of the uniporter to be identified, in part

[1]Department of Pharmacology, University of Washington, Seattle, WA, USA   [2]Howard Hughes Medical Institute and Department of Molecular Biology, Massachusetts General Hospital, Boston, MA, USA   [3]Broad Institute, Cambridge, MA, USA   [4]Department of Systems Biology, Harvard Medical School, Boston, MA, USA

Correspondence: sancak@uw.edu; vamsi@hms.harvard.edu
*Melissa JS MacEwen and Andrew L Markhard contributed equally to this work

because of its curious evolutionary distribution (Sancak et al, 2013). MCU and MICU1 homologs tend to be found in all major eukaryotic taxa, with lineage-specific losses (Bick et al, 2012). *Saccharomyces cerevisiae*, for example, has completely lost both MICU1 and MCU, and in fact, this evolutionary diversity formed the basis for the initial discovery of MICU1 (Perocchi et al, 2010). After the initial molecular identification of the uniporter machinery, our efforts to functionally reconstitute uniporter activity in yeast mitochondria using human MCU alone failed, for reasons that were not clear. This led to the search for additional missing components of the uniporter complex, leading to the identification of EMRE, which is lacking in most fungi but present in all metazoans and in extant members of the out-group of metazoans and fungi (Sancak et al, 2013). In these species, EMRE fulfills two important functions. First, the C-terminal domain of EMRE is crucial for MCU–MICU1 interaction (Sancak et al, 2013; Tsai et al, 2016). Second, EMRE is strictly required for mitochondrial Ca$^{2+}$ uptake (Sancak et al, 2013; Kovacs-Bogdan et al, 2014; Tsai et al, 2016). Hence, in metazoans, MCU and EMRE are both necessary and sufficient for reconstituting the pore activity of the uniporter.

Here, we exploited the evolutionary divergence of EMRE to understand its role in the uniporter complex. We previously showed that in amoeba *Dictyostelium discoideum*, there are no EMRE homologs, and *D. discoideum* MCU (DdMCU) forms a functional uniporter by itself (Kovacs-Bogdan et al, 2014). We reasoned that sequence elements that confer EMRE-independent activity to DdMCU could be swapped from DdMCU to *Homo sapiens* MCU (HsMCU) to render it "EMRE independent." To this end, we systematically generated HsMCU–DdMCU chimeric proteins and tested their ability to conduct Ca$^{2+}$ in human cells lacking EMRE. These efforts led to the identification of a 10-amino acid–long region in HsMCU that determines its EMRE dependence. We call this region of MCU its EMRE dependence domain (EDD). Using copper-mediated cysteine cross-linking experiments, we show that EMRE interacts with both transmembrane domains of MCU (TM1 and TM2) as well as its EDD. Interestingly, EDD, which is C-terminal to the pore-forming TM2, appears flexible in published high-resolution fungal MCU structures (Fan et al, 2018; Nguyen et al, 2018; Yoo et al, 2018) and partially overlaps with the EMRE-MCU interaction domain identified in a high-resolution cryo-EM structure of human MCU-EMRE (Wang et al, 2019). Our data suggest that EMRE stabilizes this region through direct binding, which may lead to an open conformation of the pore at the matrix side to enable exit of Ca$^{2+}$, consistent with recently reported structural data (Wang et al, 2019).

## Results

### Carboxyl-terminal domain of EMRE faces the IMS and mediates MICU1–EMRE interaction

EMRE is a small transmembrane protein that resides in the IMM and has been shown to have two distinct functions in the uniporter. First, EMRE facilitates the interaction of MCU with MICU1. Second, it is required for Ca$^{2+}$ conductance through human MCU. It was essential

to clarify EMRE's membrane topology to understand the mechanisms of these two functions. Previous experimental studies reported contradictory results on the topology of EMRE (Tsai et al, 2016; Vais et al, 2016; Yamamoto et al, 2016). We wanted to determine EMRE's topology using two complementary methods. To this end, we first generated EMRE KO cell lines that stably express EMRE protein tagged with FLAG at its carboxyl terminus (C-terminus) (EMRE-FLAG). When expressed at near endogenous levels, EMRE-FLAG rescued the mitochondrial Ca$^{2+}$ uptake defect observed in EMRE KO cells to the same extent as untagged EMRE protein (Fig 1A) in a permeabilized cell mitochondrial Ca$^{2+}$ uptake assay. Furthermore, EMRE-FLAG immunoprecipitated endogenous MCU and MICU1 (Fig 1B), showing that the C-terminal tag did not perturb EMRE's function or interaction with other uniporter proteins.

To determine the membrane topology of EMRE, we first used a proteinase accessibility assay. Mitochondria isolated from EMRE-FLAG expressing EMRE KO cells were incubated with proteinase K (PK) in the presence of increasing concentrations of digitonin, and degradation of EMRE-FLAG was monitored using Western blotting. The FLAG tag disappeared at the same digitonin concentration as IMS protein TIMM23, suggesting that the C terminus of EMRE faces the IMS (Fig 1C). Next, we confirmed N-in C-out topology of EMRE by using an orthogonal approach that uses the addition of a 5-kD mass to cysteine residues using polyethylene glycol (PEG)-maleimide (PEG5K). In this assay, cysteine residues that are in the matrix are shielded from membrane-impermeable PEG5K. Wild-type EMRE does not contain any cysteines, so we mutated S53 or S64—amino acids that are N-terminal to the predicted transmembrane domain (aa 65–84)—to cysteine (S53C or S64C). We also added a cysteine residue at EMRE's C terminus after the last amino acid (EMRE 108C). Expression of WT, S53C, S64C, or 108C EMRE in EMRE KO cells rescued the mitochondrial Ca$^{2+}$ uptake defect of these cells (Fig S1), suggesting that these mutations do not perturb protein function and topology. Mitoplasts (mitochondria without an outer membrane) prepared from EMRE KO cells that express wild type, S53C, S63C, or 108C EMRE were treated with PEG5K-maleimide. An ~5-kD shift in the molecular weight of EMRE was detected only with the EMRE 108C protein, suggesting that 53C and 64C are in the matrix. When PEG5K was added in the presence of a small amount of detergent to disrupt the inner membrane, all three cysteine-containing proteins were PEGylated, showing that the lack of modification of 53C and 64C was not due to their inaccessibility to PEG5K-maleimide in the complex (Fig 1D). These findings are consistent with previous results and confirm that EMRE's N terminus faces the matrix and its C terminus acidic domain (CAD) faces the IMM (Fig 1E) (Sancak et al, 2013; Tsai et al, 2016; Yamamoto et al, 2016).

CAD has a high percentage of negatively charged aspartic acid (D) and glutamic acid (E) (10/22 amino acids, ~45%). Notably, the presence of five or more D or E at the end of the protein is conserved across species and is a defining feature of EMRE (Sancak et al, 2013). CAD has been shown to be important for the interaction of EMRE with MICU1. We asked whether EMRE–MICU1 interaction is mainly mediated by the negative charges in this region and whether CAD also plays a role in Ca$^{2+}$ conductance. To test these, we mutated the six Ds (D102 to D107) to alanine (A) and expressed these mutant proteins in EMRE KO cells as FLAG-tagged proteins. Loss of negative

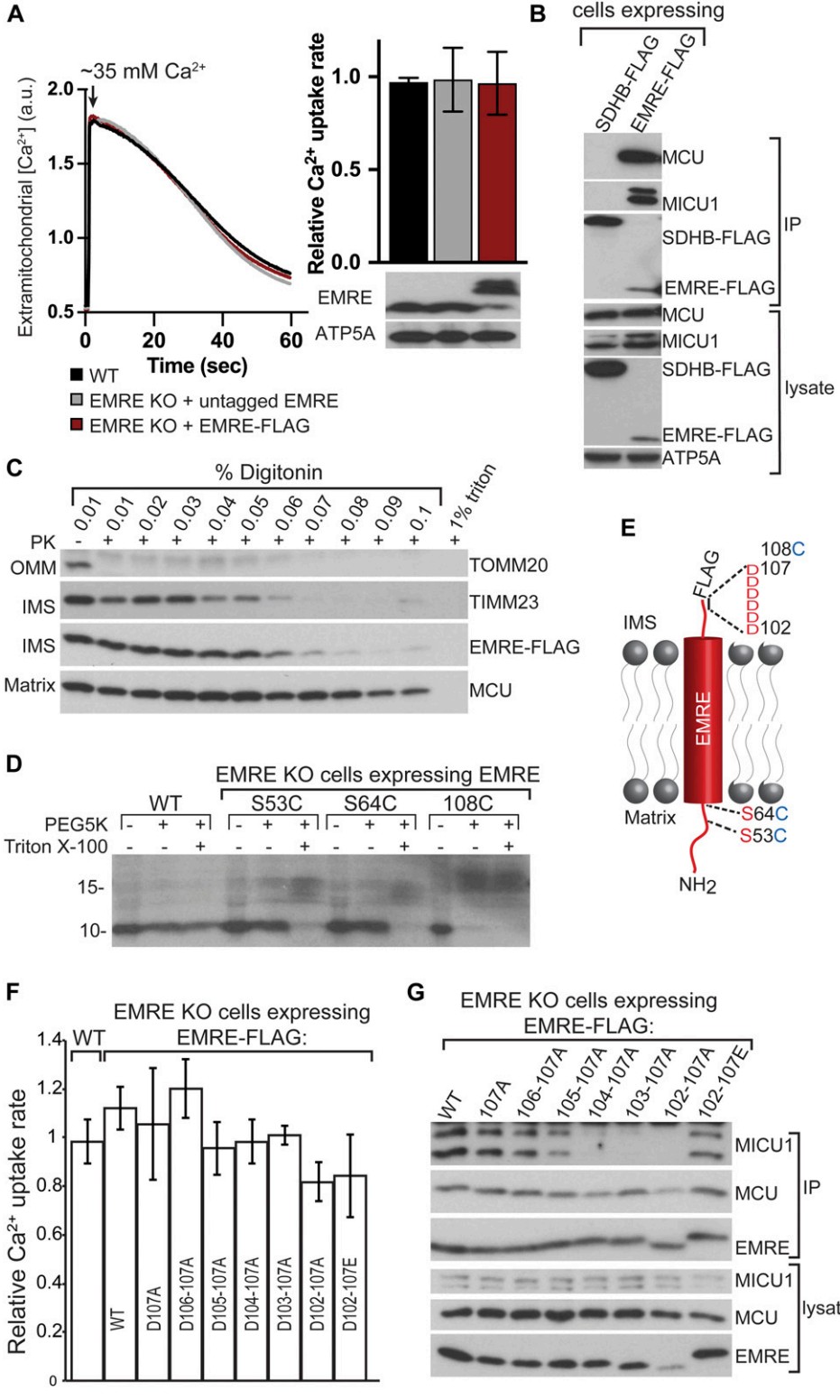

**Figure 1. EMRE CAD faces the intermembrane space and mediates EMRE–MICU1 interaction.**
**(A)** Tagging EMRE with a FLAG epitope tag at its C terminus does not impair its function. HEK293T cells expressing indicated proteins were permeabilized and mitochondrial $Ca^{2+}$ uptake was measured by monitoring extramitochondrial $Ca^{2+}$ clearance. Bar graph shows quantification of $Ca^{2+}$ uptake rates and Western blot shows EMRE expression. (n = 4). ATP5A serves as loading control. **(B)** C-terminal FLAG tag does not impair EMRE–MCU and EMRE–MICU1 interactions. EMRE FLAG and control SDHB-FLAG were immunoprecipitated, and immunoprecipitates were blotted for MCU and MICU1. ATP5A serves as loading control. **(C)** Proteinase K treatment of isolated mitochondria in the presence of increased detergent concentration. EMRE-FLAG is degraded by proteinase K at the same detergent concentration as TIMM23, an inner mitochondrial membrane protein. **(D)** Mitochondria were isolated from WT or EMRE KO cells that stably express the indicated proteins. Mitoplasts (mitochondria without outer membranes) were prepared and treated with PEG5K-maleimide. A 5-kD mass addition to EMRE protein was detected by Western blotting. **(D, E)** Schematic shows EMRE membrane topology and the position of the amino acids that were mutated to cysteines for PEGylation experiments shown in (D). EMRE aa 64–85 were predicted to form its transmembrane domain using TMHMM (Sonnhammer et al, 1998). **(F)** EMRE DDD domain is not required for mitochondrial calcium uptake. Mitochondrial calcium uptake rates of WT and EMRE KO cells stably expressing the indicated proteins (n = 4). **(G)** Charge-conserving mutations of the six aspartic acids of EMRE to glutamic acid restores EMRE–MICU1 interaction. WT EMRE–FLAG or EMRE–FLAG with the indicated mutations were stably expressed in EMRE KO cells, immunoprecipitated, and immunoprecipitates were subjected to Western blotting to detect EMRE–MICU1 interaction. Data information: In (A, F), data are presented as mean ± SD.

charge in this region did not perturb mitochondrial $Ca^{2+}$ uptake (Fig 1F) (Tsai et al, 2016; Vais et al, 2016; Yamamoto et al, 2016). However, we did see a decrease in the amount of MICU1 that immunoprecipitated with EMRE as the number of alanines increased in this region. EMRE–MICU1 interaction was restored when Ds were mutated to similarly charged Es (Fig 1G). These results show that the negative charge of CAD is dispensable for $Ca^{2+}$ conductance by MCU but is critical for EMRE–MICU1 interaction.

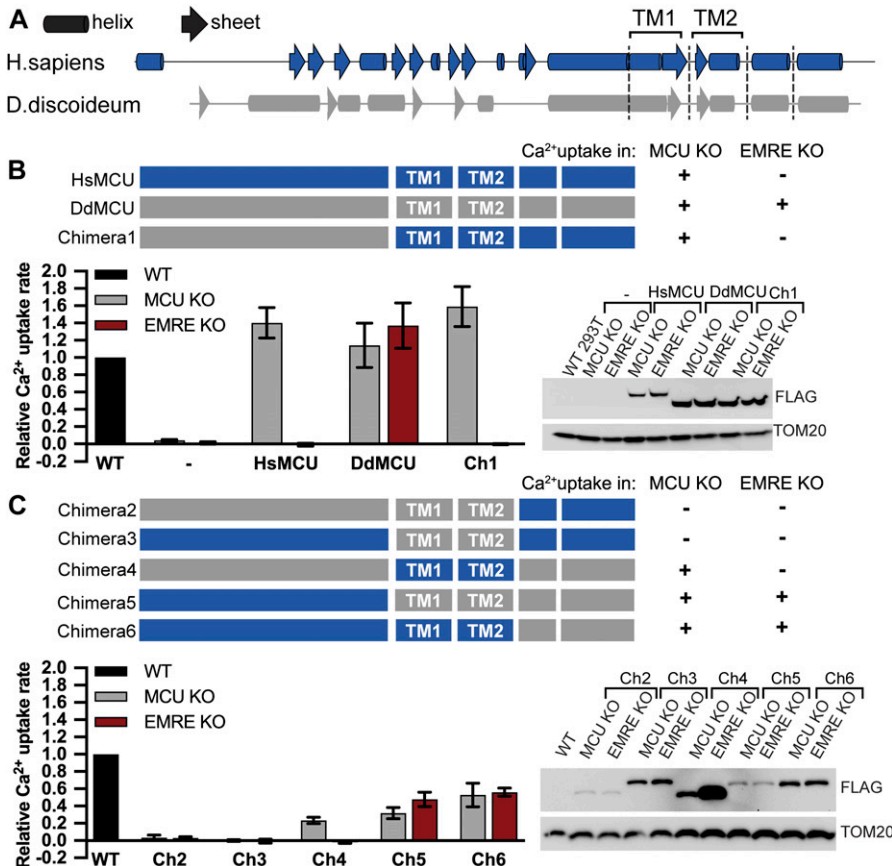

**Figure 2. Carboxyl-terminal helices of *Dictyostelium discoideum* MCU (DdMCU) confer EMRE-independent Ca²⁺ uptake to *Homo sapiens* MCU (HsMCU).**

**(A)** Schematic shows helices and sheets of HsMCU and DdMCU as predicted by PSIPRED. Two transmembrane domains are labeled. **(B, C)** Schematic summarizes the domain structure of HsMCU, DdMCU, and the chimeric proteins. FLAG-tagged proteins were stably expressed in MCU KO and EMRE KO HEK293T cells. Mitochondrial Ca²⁺ uptake rates in control WT and chimera expressing cells were measured and normalized to those of WT cells (n = 3–4). Expression of chimeras was detected by Western blotting using anti-FLAG antibody. TOM20 serves as loading control. Data information: in (B, C), data are presented as mean ± SD.

## Identification of the EDD of HsMCU

EMRE likely arose in early evolution of opisthokonts and is found in all metazoans. Its loss leads to a complete loss of mitochondrial Ca²⁺ uptake (Sancak et al, 2013), but the molecular basis for this requirement is not known. We previously showed that amoeba *D. discoideum* does not have an EMRE homolog and that *D. discoideum* MCU (DdMCU) forms a functional Ca²⁺ channel by itself (Kovacs-Bogdan et al, 2014). In contrast, to be able to conduct Ca²⁺, human MCU (HsMCU) requires co-expression of EMRE (Kovacs-Bogdan et al, 2014). The EMRE dependence of MCU does not appear to be related to proper MCU folding or mitochondrial localization, as MCU forms higher order oligomers with correct membrane topology in the absence of EMRE (Kovacs-Bogdan et al, 2014). We hypothesized that EMRE plays an important role in Ca²⁺ permeation of the human uniporter and exploited the evolutionary divergence of EMRE dependence of DdMCU and HsMCU to understand the molecular details of EMRE function. First, we compared the predicted secondary structures of DdMCU and HsMCU (Fig 2A). The predicted secondary structures of the two MCU proteins were most divergent at their N termini. To test whether the DdMCU N terminus domain would be sufficient to confer EMRE independence to HsMCU, we generated a chimeric protein, chimera 1, as shown in Fig 2B. We expressed chimera 1 in MCU KO HEK293T cells to determine whether it would form a functional protein, and in EMRE KO HEK293T cells to determine whether it would function independently of EMRE

in our permeabilized cell mitochondrial Ca²⁺ uptake assay. Chimera 1 formed a functional channel in MCU KO cells; however, its activity was still dependent on the presence of EMRE (Fig 2B).

These results suggested that the two transmembrane domains or the C terminus domain of HsMCU might be critical for its EMRE dependence. To test this, we generated chimeras 2–6 and determined their function and EMRE dependence. Chimeras 2 and 3 did not form functional Ca²⁺ channels (Fig 2C). Chimera 4 showed reduced, but EMRE-dependent uniporter activity (Fig 2C), suggesting that the TM domains of HsMCU are involved in its EMRE dependence. Chimeras 5 and 6, on the other hand, showed EMRE-independent Ca²⁺ uptake (Fig 2C). Both of these chimeras contain the two predicted C-terminal helices from DdMCU. To determine whether one of these predicted helices might be the critical domain for EMRE-independent Ca²⁺ uptake, we generated chimeras 7 and 8. Surprisingly, chimera 7 was functional to the same extent both in MCU KO and EMRE KO cells. In addition, chimera 8, despite its poor expression, supported mitochondrial Ca²⁺ uptake, and its activity was EMRE dependent (Fig 3A). The helical region that defines chimera 7 is composed of 23 amino acids. We further divided this region into two halves at conserved amino acids to generate chimeras 9 and 10 (Figs 3B and S2). Chimera 10 did not support mitochondrial Ca²⁺ uptake, but chimera 9 formed an EMRE-independent Ca²⁺ channel (Fig 3B). The chimera 9 region originating from DdMCU is 10 amino acids long and is located directly after the pore forming TM (TM2) of HsMCU. We term this 10-amino acid region of MCU (aa288–aa297) the EDD (Fig 3C).

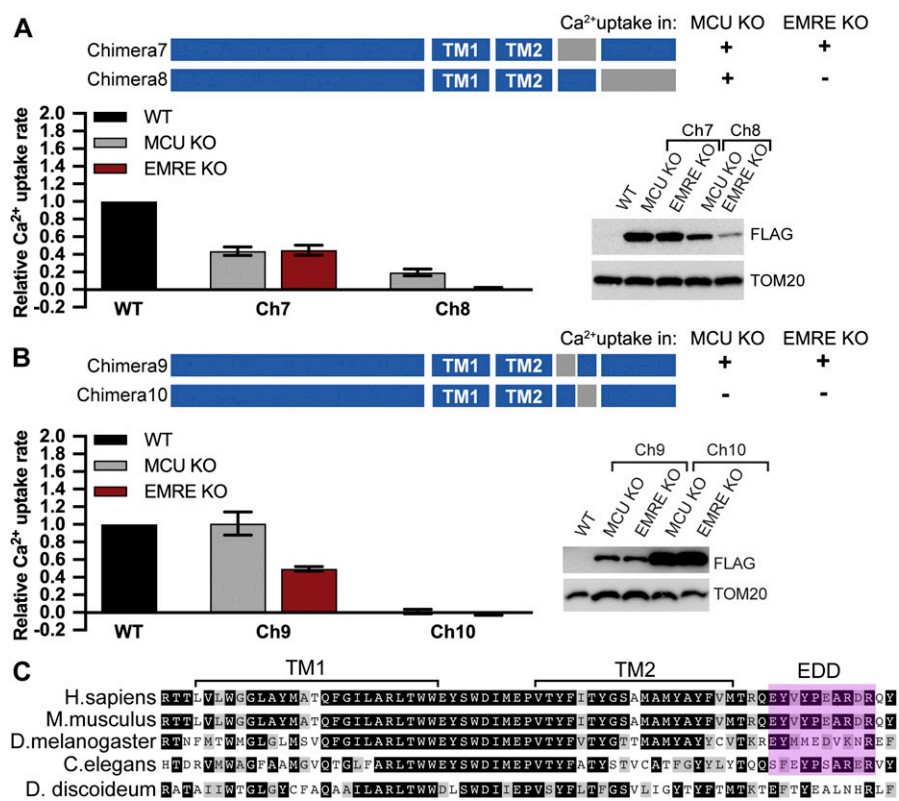

**Figure 3. EMRE dependence domain of *Homo sapiens* MCU is a 10- amino acid–long region located C-terminal to TM2.**
**(A, B)** Schematic summarizes the domain structure of chimeric proteins. FLAG-tagged proteins were stably expressed in mitochondrial calcium uniporter (MCU) KO and EMRE KO HEK293T cells. Mitochondrial Ca²⁺ uptake rates in control WT and chimera expressing cells were measured and normalized to those of WT cells (n = 3–4). **(C)** Alignment of MCU protein from indicated species was done using CLUSTALW and amino acids were color-coded using BoxShade. Black boxes show identical amino acids, gray boxes show similar amino acids. TM1, TM2 and EMRE dependence domain are indicated. Data information: In (A, B), data are presented as mean ± SD.

During our analysis, we noticed that the expression levels of the chimeric proteins varied (Figs 2B and C, 3A and B, and S3A). However, we did not observe a correlation between the expression level of a particular chimera and the rate of mitochondrial Ca²⁺ uptake in cells that express the chimeric proteins (for example, compare the chimera 4 and 5 in Fig 2C). In these experiments, chimeric proteins were stably expressed in cells using lentivirus-mediated integration of the corresponding cDNAs into the genome. To eliminate the possibility that low protein expression was due to low virus titer, we picked chimera 5, a chimera that expressed poorly but formed functional channels, and reinfected cells with virus. We observed a slight increase in protein expression and a concomitant increase in Ca²⁺ uptake rates. However, chimera 5 still expressed at lower levels than HsMCU (Fig S3B). This suggests that chimeric proteins may be inherently unstable. Consequently, we cannot determine whether low Ca²⁺ uptake rates of chimeras compared to HsMCU are due to their channel properties or their expression levels. Nevertheless, expression of a particular chimeric protein in MCU KO and EMRE KO cells were mostly comparable, allowing us to determine their EMRE dependence. Expression of HsMCU–DdMCU chimeras did not alter mitochondrial membrane potential, suggesting that lack of mitochondrial Ca²⁺ uptake after expression of some chimeras was not secondary to perturbed mitochondrial health (Fig S4). Example mitochondrial Ca²⁺ uptake data for HsMCU, DdMCU, and chimeras in MCU KO and EMRE KO cells are shown in Fig S5 and sequences of the chimeric proteins used are provided (Supplemental Data 1).

## MCU TM1, TM2, and EDD interact with EMRE

Our functional experiments highlighted the importance of MCU TM domains and EDD for the EMRE dependence of human MCU. To complement these experiments and to determine if these domains are also important for EMRE–MCU physical interaction, we performed immunoprecipitation experiments using chimeras that formed functional channels and showed good protein expression (chimeras 1, 5, 6, 7, and 9). To further normalize protein expression levels, we transiently expressed control HsMCU, DdMCU, or chimeras, together with untagged EMRE in MCU KO cells. We treated these cells with amine-reactive cross-linker dithiobis (succinimidyl propionate) (DSP) before cell lysis to stabilize protein–protein interactions. We then immunoprecipitated MCU-FLAG. Chimera 5 did not interact with EMRE; chimeras 6 and 7 showed reduced EMRE interaction; chimeras 1 and 9 showed wild-type levels of EMRE–MCU interaction (Fig 4A and B). Consistent with stabilization of EMRE protein when bound to MCU, chimeras that showed better EMRE interaction also had more EMRE protein in the lysate (Fig 4A). These experiments suggested that MCU TM1, TM2, and the helical region that includes the EDD are important for EMRE–MCU binding. To determine if EMRE directly interacts with HsMCU in TM1, TM2, and EDD, we performed cysteine cross-linking experiments. Two amino acids on each transmembrane domain of MCU were selected and mutated to cysteine. We also mutated several consecutive EMRE amino acids to cysteines and performed a partial cysteine-scanning experiment.

First, we co-expressed MCU and EMRE proteins that contain one cysteine residue each, as well as control cysteine-free MCU, in MCU

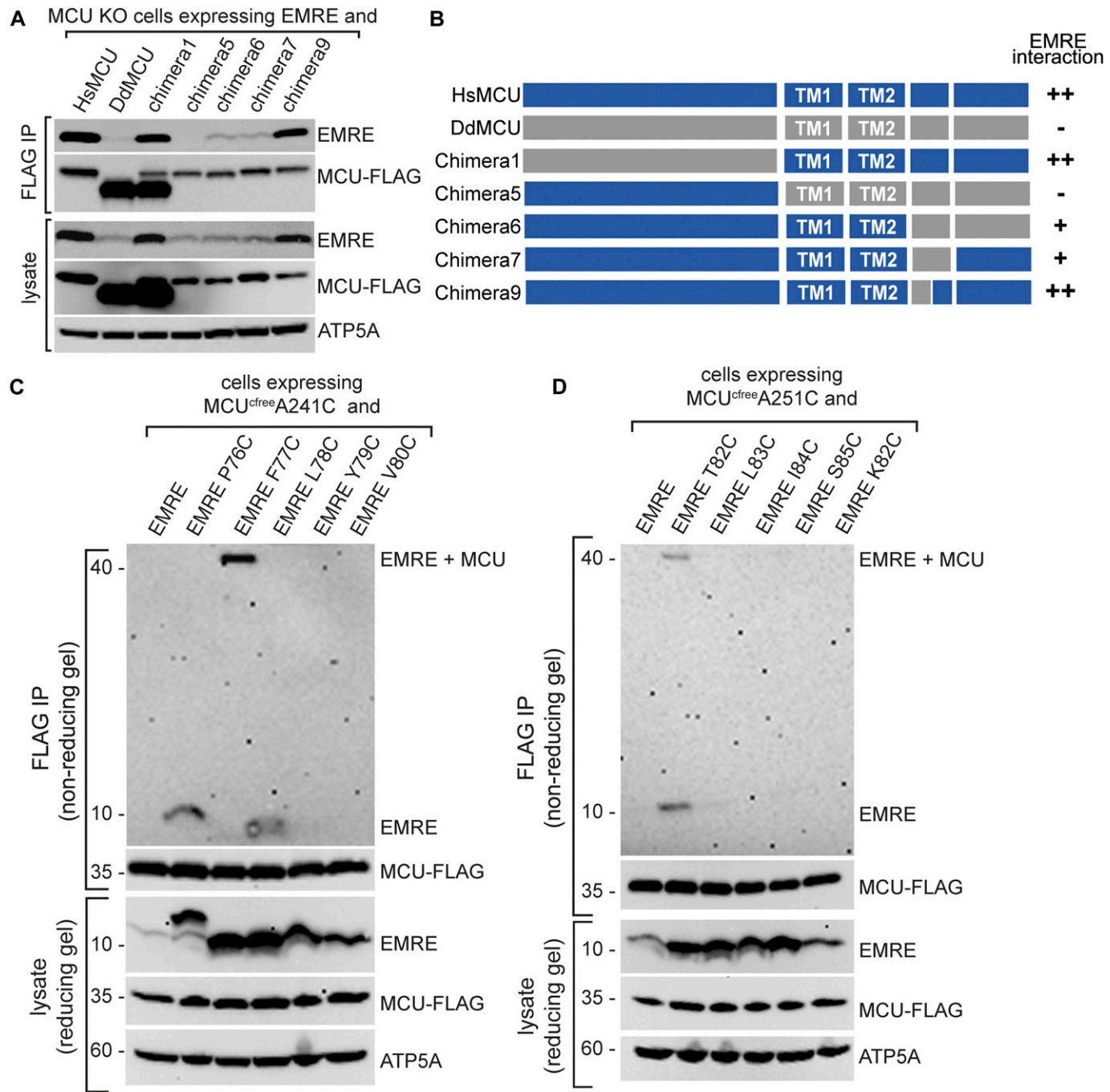

**Figure 4. EMRE directly interacts with TM1 of mitochondrial calcium uniporter (MCU).**
**(A)** MCU TM and C-terminal helices are required for EMRE–MCU interaction. Untagged EMRE and indicated FLAG-tagged MCU proteins were co-expressed in MCU KO HEK293T cells by transient transfection, FLAG-tagged proteins were immunoprecipitated and immunoprecipitates were analyzed for the presence of EMRE by Western blotting. ATP5A serves as loading control. EMRE–MCU interaction was evident both in immunoprecipitates and in lysates through stabilization of EMRE. **(B)** Schematic summarizes EMRE-chimera binding data and highlights the importance of MCU TM and C-terminal helices for EMR–EMCU interaction. **(C, D)** EMRE–*Homo sapiens* MCU cysteine cross-linking experiments show direct binding of MCU TM1 residues A241 and A251 to EMRE F77 and T82, respectively. **(C, D)** *H. sapiens* MCU that contains only one cysteine at amino acid 241 (C) or 251 (D) were stably co-expressed with indicated EMRE proteins. WT EMRE does not contain any cysteines and served as a control. Mitochondria were isolated from cells, and cysteine–cysteine cross-linking was induced using copper phenanthroline. MCU-FLAG was immunoprecipitated and the presence of an ~40 kD cross-linked EMRE-MCU band was detected under non-reducing conditions by Western blotting. Lysates were prepared in parallel under reducing conditions and were blotted to detect indicated proteins. ATP5A serves as loading control. Numbers indicate the locations of molecular weight standards.

KO cells and confirmed that they are functional (Fig S6A). WT EMRE does not contain any cysteine residues and served as an additional control. Then, we performed copper-mediated cysteine cross-linking experiments in mitochondria isolated from these cells, immunoprecipitated MCU, and determined the presence of EMRE–MCU cross-linked protein product using non-reducing gel electrophoresis followed by Western blotting. Our results showed that MCU TM1 residue A241C cross-links with EMRE F77C, but not with four other EMRE residues near F77C (Fig 4C). Similarly, MCU TM1 residue A251C specifically cross-linked to EMRE T82C (Fig 4D). To determine if EMRE also interacts with the pore-forming TM2 of MCU, we performed similar cross-linking experiments with MCU I270C and MCU M276C. MCU KO cells expressing these MCU and EMRE cysteine cross-linked residues showed $Ca^{2+}$ uptake, showing that cysteine substitutions did not alter protein function (Fig S6B). MCU I270C cross-linked to EMRE I84C (Fig 5A) and MCU M276C cross-linked to EMRE P76C (Fig 5B). Collectively, these findings confirm the interaction that was observed between MCU TM1 and EMRE previously (Tsai et al, 2016) and establish that the pore-forming TM2 of MCU also interacts with EMRE, both within and outside the MCU–EMRE interaction domain defined by the MCU–EMRE structure (Wang et al, 2019).

Finally, we performed cysteine cross-linking experiments between amino acids in EDD (MCU E293 and D296) and the N-terminal domain of EMRE that faces the mitochondrial matrix (K59 and K62). First, we confirmed that cysteine substitutions did not alter the function of MCU or EMRE (Fig S6C). Both MCU residues in EDD cross-linked to both EMRE residues (Fig 5C). Fig 5D shows a schematic of MCU and EMRE amino acids that cross-linked. To gain insight for functional significance of EDD for uniporter function, we identified and highlighted the sequences that correspond to EDD in the four fungal MCU homologs whose high-resolution structures have been published (Fig S7). Surprisingly, this region appeared flexible in fungal MCU, which have slow calcium conductance rates compared to human uniporter (Carafoli & Lehninger, 1971; Goncalves et al, 2015; Fan et al, 2018; Nguyen et al, 2018; Wettmarshausen et al, 2018; Pittis et al, 2020 Preprint). Based on this observation and our cross-linking data, we posit that binding of EMRE stabilizes this otherwise flexible region at the matrix opening of the channel and allows exit of $Ca^{2+}$ ions from the pore.

Previous studies suggested that a small portion of N-terminal domain of EMRE is dispensable for MCU–EMRE interaction (Tsai et al, 2016). However, in contrast, our cross-linking data show that the N terminus of EMRE directly interacts with MCU. To determine the importance of this region for uniporter function and the stability of EMRE–MCU interaction, we generated chimeric proteins using HsMCU, HsEMRE, Caenorhabditis elegans MCU, and C. elegans EMRE (CeEMRE) as shown in Fig 5E. When expressed in MCU KO cells, C. elegans MCU and CeEMRE form a functional channel, but HsMCU and CeEMRE are not compatible (Tsai et al, 2016). Sequence alignment of MCU and EMRE from human and C. elegans are shown in Fig S8. This system allowed us to test whether the EDD and N terminus of EMRE contribute to MCU–EMRE interaction and channel function by generating chimeric C. elegans and human proteins. HsEMRE with CeEMRE N terminal domain (HsEMRE[CeNterm]) did not bind to HsMCU. When HsEMRE[CeNterm] was expressed with HsMCU with CeEDD (HsMCU[CeEDD]), the two proteins interacted and

supported mitochondrial $Ca^{2+}$ uptake (Fig 5F and G). We conclude that EDD and EMRE N terminus interactions are necessary to form a stable association between MCU and EMRE and to form a functional pore.

Based on our functional and interaction data, we propose a mechanism of EMRE regulation of the mitochondrial $Ca^{2+}$ channel that is consistent with high-resolution MCU-EMRE structure (Wang et al, 2019) and previous functional studies (Sancak et al, 2013): when MCU is expressed in the absence of EMRE, there is no calcium transport, likely because the EDD is flexible and prevents $Ca^{2+}$ exit from the pore. Co-expression of MCU and EMRE are necessary and sufficient for calcium transport (Kovacs-Bogdan et al, 2014), likely because EMRE binds to the transmembrane domains and EDD of MCU, changes the EDD conformation, permitting exit of $Ca^{2+}$ ions (Fig 5H).

## Discussion

Perturbation of uniporter function is associated with a number of cellular and systemic defects, ranging from altered cell cycle progression and mitochondrial dynamics to skeletal muscle myopathy and neurodegenerative disease (Kamer & Mootha, 2015; Chakraborty et al, 2017; Musa et al, 2019). EMRE has emerged as a core component of the animal mitochondrial $Ca^{2+}$ uniporters whose expression is under transcriptional and posttranslational control (Konig et al, 2016; Munch & Harper, 2016; Tsai et al, 2017). For example, accumulation of EMRE protein in the absence of mitochondrial AAA-proteases AFG3L2 and SPG7, whose mutations are associated with spinocerebellar ataxia and hereditary spastic paraplegia, is responsible for mitochondrial $Ca^{2+}$ overload and may contribute to neuronal loss (Konig et al, 2016). In addition, in a mouse model of neuromuscular disease caused by MICU1 deficiency, decreased EMRE expression over time correlated with improved health (Liu et al, 2016). These observations highlight the importance of EMRE in physiology and disease.

Here, we exploited evolutionary divergence of mitochondrial $Ca^{2+}$ uniporter composition to understand the function of EMRE, which is required for the human uniporter but not found in most fungi or other taxa. Functional experiments using chimeric proteins that consist of human HsMCU (which is EMRE dependent) and Dictyostelium DdMCU (which operates independent of EMRE) revealed the presence of a region in MCU that we named EDD. We also show that EMRE makes direct contacts with the two TM domains of MCU as well as with EDD. Interestingly, the region that corresponds to EDD appears flexible in previously published high-resolution structures of fungal MCU homologs (Baradaran et al, 2018; Fan et al, 2018; Nguyen et al, 2018) and partially overlaps with the juxtamembrane loop identified to be important to stabilize channel opening in structural studies (Wang et al, 2019). In species that do not have EMRE, it is plausible that lipids or other currently unknown proteins may fulfill the same function. It is notable that fungal MCU homologs appear to have extremely low conductance, as initially documented by Lehninger and colleagues (Carafoli & Lehninger, 1971) and later by others (Goncalves et al, 2015; Fan et al, 2018; Nguyen et al, 2018; Wettmarshausen et al, 2018; Pittis et al, 2020

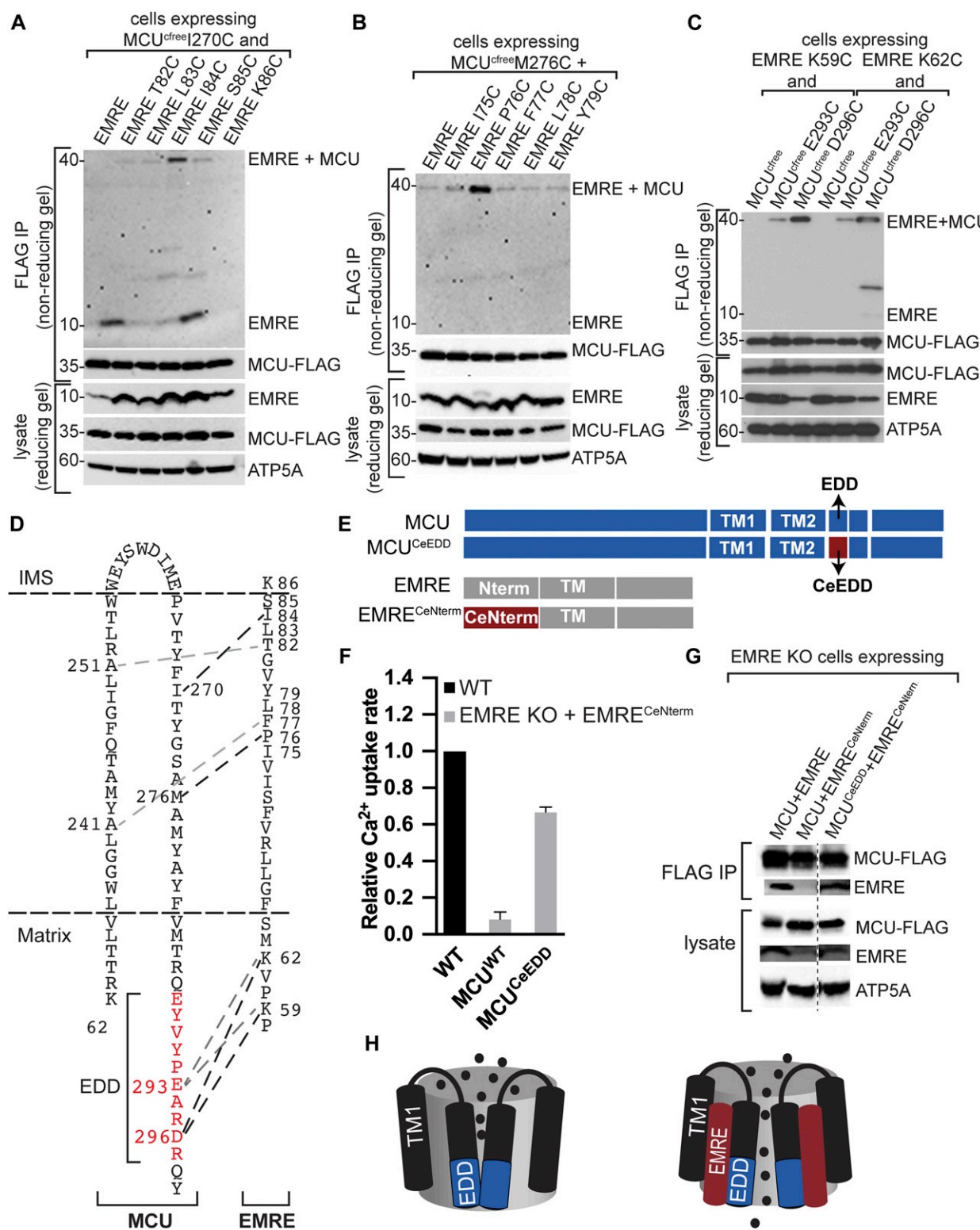

**Figure 5. EMRE directly interacts with TM2 and EMRE dependence domain (EDD) of mitochondrial calcium uniporter (MCU).**
**(A, B)** EMRE–*Homo sapiens* MCU (HsMCU) cysteine cross-linking experiment shows direct binding of MCU TM2 residues I270 and M276C to EMRE I84 and P76, respectively.
**(A, B)** HsMCU that contains only one cysteine at amino acid 270 (A) 276 (B) was stably co-expressed with indicated EMRE proteins in MCU KO cells. WT EMRE does not contain any cysteines and served as a control. Mitochondria were isolated from cells and cysteine–cysteine cross-linking was induced using copper phenanthroline. MCU-FLAG was immunoprecipitated and the presence of an ~40 kD cross-linked EMRE–MCU band was detected under non-reducing conditions by Western blotting using EMRE antibody. Lysates were prepared in parallel under reducing conditions and were blotted to detect indicated proteins. ATP5A serves as loading control. Numbers

*Preprint*). In contrast, we have shown that the uniporter of animals and those from *Dictyostelium* exhibit extremely high conductance (Kovacs-Bogdan et al, 2014).

In our experimental system, the measured activity of a chimeric protein is a function of its expression, channel properties, and its effect on mitochondrial membrane potential and mitochondrial health. We did not observe perturbed membrane potential in cells that express chimeras. However, it is possible that some chimeras exhibit reduced mitochondrial uptake compared with HsMCU or DdMCU because their stable expression causes mitochondrial Ca$^{2+}$ overload or alters mitochondrial Ca$^{2+}$ storage capacity or Ca$^{2+}$ extrusion from the mitochondria. Despite having variable expression levels of HsMCU, DdMCU, and chimeras, expression of a particular chimera was comparable in MCU KO and EMRE KO cells, with the exception of chimera 4. This enables us to determine EMRE dependence of a chimera by comparing Ca$^{2+}$ uptake rates in MCU KO and EMRE KO cells, independent of how its expression compares with that of HsMCU.

Our data show that EMRE directly interacts with MCU in TM1, TM2, and EDD. Importantly, comparison of chimera 7 and chimera 9 immunoprecipitation data (Fig 4A) suggests that the region defined by chimera 10 might also be important for EMRE–MCU binding. However, chimera 10 was not functional, and we did not pursue this chimera for protein–protein interaction experiments because of the possibility that it is not folded properly. In addition, although EDD is the smallest region that we tested that enabled EMRE-independent Ca$^{2+}$ uptake, it is evident in our data that the presence of EMRE can increase Ca$^{2+}$ uptake rates by chimera 9. Thus, we note that the data presented here are consistent with additional, extended interactions between EMRE and MCU in the matrix.

Notably, our functional data and model for the role of EMRE in Ca$^{2+}$ conductance appear to be generally in good agreement with a high-resolution structure of MCU–EMRE that was contemporary with the preprint version of this article (Wang et al, 2019). Wang et al (2019) also conclude that EMRE enables Ca$^{2+}$ conductance by modulating the conformation of MCU distal to the Ca$^{2+}$ pore. Both studies have also converged on an overlapping region of MCU that confers EMRE dependence: Wang et al (2019) spotlighted a six–amino acid–long region (aa 285–aa 291) in HsMCU, whereas we defined EDD as MCU aa 288–aa 297 using evolutionary divergence and systematic domain swapping. Future studies will determine whether addition of aa 285–aa 287 to EDD will augment its activity. In addition, both our experimental data and those published by Wang et al (2019) show that EMRE interacts with MCU TM1 and TM2. However, our data show robust cross-linking between TM2 and EMRE amino acids that appear farther apart than cross-linking

distance in the structure. At present, we cannot reconcile this experimental finding with the structure. Although the cross-linking could be spurious, our results are robust and raise the possibility that there may be additional conformational states of the complex that are not reflected in the structure, or cysteine cross-linking may be capturing dynamics that occur in the complex's native environment (Bass et al, 2007).

What is the evolutionary significance of EMRE? Curiously, based on functional data, *Dictyostelium* and fungal MCU appear to be able to adopt an open conformation when expressed (Kovacs-Bogdan et al, 2014; Baradaran et al, 2018; Fan et al, 2018; Nguyen et al, 2018; Yoo et al, 2018), whereas human MCU when expressed on its own adopts a closed conformation. *Dictyostelium* MCU displays a much higher rate of calcium uptake than those from fungi and comparable with human MCU and EMRE (Kovacs-Bogdan et al, 2014). The current work suggests that the EDD of MCU is responsible for maintaining the closed state, but in a manner that is dependent on EMRE. Given that MICU1/2 interacts with EMRE, it is conceivable that EMRE mediates acute allosteric control at the exit site. Over longer time scales, the expression level of EMRE has emerged as an important determinant of uniporter activity (Konig et al, 2016; Tsai et al, 2017) with relevance to human disease. Collectively, these findings suggest that this evolutionary innovation may have emerged to confer an added layer of acute or chronic regulation to the uniporter. Future structural and functional studies will be required to fully decipher the mechanisms by which the uniporter is regulated across different eukaryotes.

# Materials and Methods

## Cell culture

- DMEM; Thermo Fisher Scientific, Cat. no. 11-965-118
- FBS; Life Technologies, Cat. no. 26140087
- GlutaMAX; Thermo Fisher Scientific, Cat. no. 35-050-061
- Trypsin; Gibco, Cat. no. 12605-010
- PBS; Thermo Fisher Scientific, Cat. no. 20012050
- Penicillin/streptomycin solution; VWR Cat. no. 45000-652
- Genlantis MycoScope PCR Detection Kit; VWR Cat. no. 10497-508

HEK293T cells were acquired from the Sabatini Lab at the Whitehead Institute for Biomedical Research. They were grown in DMEM medium supplemented with 1× GlutaMAX and 10% FBS. The cells were tested for mycoplasma every 3 mo using the Genlantis MycoScope PCR Detection Kit and were confirmed to be free of

---

indicate the locations of molecular weight standards. **(C)** EMRE–HsMCU cysteine cross-linking experiments show direct binding of MCU EDD residues E293 and D296 to EMRE K59 and K62. **(A)** Cross-linking and sample processing were performed as in (A). **(D)** Schematic showing MCU and EMRE amino acids that directly interact with each other in the membrane and in the matrix. EDD is shown in red. **(E)** Schematic showing HsMCU, HsMCU with *Caenorhabditis elegans* MCU EDD, Hs EMRE, and HsEMRE with *C. elegans* EMRE N-terminal domain. **(F, G)** These constructs were used in (F, G). **(F)** Mitochondrial Ca$^{2+}$ uptake rates in control WT or EMRE KO cells stably expressing HsEMRE$^{CeNterm}$ together with HsMCU or HsMCU$^{CeEDD}$ were measured and normalized to those of WT cells (n = 3–4). MCU forms a functional channel only if its EDD interacts with EMRE. **(G)** MCU–FLAG was immunoprecipitated from EMRE KO cells that stably express HsMCU or HsMCU$^{CeEDD}$ with HsEMRE or HsEMRE$^{CeNterm}$ after DSP-mediated cross-linking. Immunoprecipitates and lysates were analyzed with Western blotting for the presence of indicated proteins. An interaction with MCU and EMRE was observed only if EDD and EMRE originated from the same species. **(H)** Model shows the proposed mechanism of EMRE function in the uniporter. In the absence of EMRE, Ca$^{2+}$ ions cannot exit the channel because of blockage of the pore by EDD. Binding of EMRE leads to a conformational change in EDD and allows exit of Ca$^{2+}$ ions into the matrix. **(F)** Data information: in (F), data are presented as mean ± SD.

mycoplasma contamination. The identity of the HEK293T cells was confirmed using short tandem repeat analysis. The HEK293T cell line has the following short tandem repeat profile: TH01 (7, 9.3); D21S11 (28, 29, 30.2); D5S818 (7, 8, 9); D13S317 (11, 12, 13, 14, 15); D7S820 (11); D16S539 (9, 13); CSF1PO (11, 12, 13); Amelogenin (X); vWA (16, 18, 19, 20); TPOX (11). This profile matches 100% to HEK293T cell line profile (CRL-3216; ATCC) if the Alternative Master's algorithm is used, and 83% if the Tanabe algorithm is used.

## Gel electrophoresis and Western blotting

### Antibodies and dilution used for experiments
- MCU antibody; Sigma-Aldrich, Cat. no. HPA016480-100UL (1:2,000)
- DYKDDDDK Tag Rabbit antibody; Cell Signaling Technology, Cat. no. 14793S (1:3,000)
- EMRE antibody; Bethyl Laboratories, Cat. no. A300-BL19208 (1:1,000)
- ATP5A antibody; Abcam Biochemicals, Cat. no. ab14748 (1:5,000)
- TOM20 antibody; Cell Signaling Technology, Cat. no. 42406S (1:5,000)
- HRP-linked antirabbit secondary antibody; Cell Signaling Technology Cat. no. 7074S (1:10,000)
- HRP-linked antimouse secondary antibody; Cell Signaling Technology Cat. no. 7076S (1:10,000)

### Gel electrophoresis
- 10× Tris/Glycine Buffer; Boston BioProducts Cat. no. BP-150-4L
- Novex WedgeWell 16% Tris-Glycine Gel; Invitrogen Cat. no. XP00165BOX
- Novex 12% Tris-Glycine Mini Gels, WedgeWell format, 15-well; Thermo Fisher Scientific Cat. no. XP00125BOX

### Western blotting
- 10× TBST-Standard (10× w/1% Tween-20, pH 7.4); Boston BioProducts Cat. no. IBB-580-4L
- Ethanol, 200 proof (100%); Thermo Fisher Scientific Cat. no. 04-355-450
- Trans-Blot Turbo 5× Transfer Buffer; Bio-Rad Cat. no. 10026938
- Powdered fat-free milk; Kroger brand, Cat. no. G500A554
- Trans-Blot Turbo RTA Mini PVDF Transfer Kit; Bio-Rad Cat. no. 1704272
- Transfer apparatus for SDS–PAGE; Bio-Rad Trans-Blot Turbo Transfer System

   For MCU-FLAG and ATP5A; Bio-Rad Mixed Molecular Weight Protein Transfer setting (7 min, 1.3 A, 25 V)
   For TOM20; Bio-Rad Low Molecular Weight Protein Transfer setting (5 min, 1.3 A, 25 V)
- Blot imager: iBrightCL1000
- Automated Western Blot Development Processor: Precision Biosystems BlotCycler, Model W5 100-12VAC; S/N 394387
- Clarity Max Western ECL Substrate; Bio-Rad Cat. no. 1705062
- Clarity Western ECL Substrate; Bio-Rad Cat. no. 170-5060

   After transfer, the membranes were briefly washed with TBST and incubated with 5% milk in TBST (wt/vol) for 30 min. They were then incubated overnight with primary antibodies diluted in 5% milk in TBST (wt/vol). Afterward, all membranes were washed with TBST three times, 5 min each, and incubated for 1 h with secondary antibody diluted in 5% milk in TBST at room temperature. The membranes were then washed four times, 5 min each, using a Precision Biosystems BlotCycler. Membranes were developed using Bio-Rad ECL substrate.

### Transfer and PFA cross-linking of EMRE blots
- Transfer apparatus for SDS–PAGE; Bio-Rad Trans-Blot Turbo Transfer System

   A custom 3-min transfer protocol with constant 1.3 A and 25 V was used for EMRE Western blotting
- 16% Paraformaldehyde aqueous solution; EMS/Thermo Fisher Scientific Cat. no. 50-980-487
- PBS; Thermo Fisher Scientific, Cat. no. 20012050

   Protocol and reagents adapted from (Suzuki et al, 2008). Immediately after transfer of proteins from the electrophoresis gel to a 0.22-$\mu$m polyvinylidene difluoride (PVDF) membrane, membranes that were to be immunoblotted for EMRE were soaked in a solution of 0.4% PFA in PBS for 30 min without agitation. The membranes were then blocked and immunoblotted normally, as described above.

### Cell lysis, sample preparation, and immunoprecipitation

- Bradford Dye Reagent; Bio-Rad, Cat. no. 5000205
- Spectrophotometer; Spectronic Instruments, Genesys 5
- Protease inhibitors; Sigma-Aldrich, Cat. no. 5892953001
- Lysis buffer reagents:
- Hepes–KOH

   Hepes; Sigma-Aldrich Cat. no. H3375-1KG
   KOH; Sigma Millipore Cat. no. 1050121000
- NaCl; Sigma-Aldrich Cat. no. 746398-5KG
- EDTA; Sigma-Aldrich, Cat. no. 607-429-00-8
- Triton X-100; Sigma-Aldrich, Cat. no. X100-1L
- DDM; Sigma-Aldrich, Cat. no. D4641-5G
- Reducing sample buffer, pH 6.8:

   SDS; Sigma-Aldrich Cat. no. L4509-1KG
   BME/2-mercaptoethanol; Sigma-Aldrich Cat. no. M3148-25ML
   Glycerol; Sigma-Aldrich Cat. no. G5516-1L
   Tris–HCl: Trizma base; Sigma-Aldrich Cat. no. RDD008
   Bromophenol Blue; VWR Cat. no. 97061-690
- Non-reducing sample buffer, pH 6.8: same as "reducing sample buffer," but without BME.

   For standard lysis, cell plates were placed on ice and washed with cold PBS, which was then aspirated. Cells were harvested in lysis buffer supplemented with 1% Triton X-100 (with 0.2% DDM if lysates were also used for immunoprecipitation) and proteases inhibitors; the volume of lysis buffer used varied based on downstream uses. Cells were triturated in tubes and then centrifuged at 17,000$g$ for 10 min. Cell supernatant was quantified using a Bradford protein assay and a spectrophotometer. Sample preparation varied based on downstream applications.

## Membrane potential measurements

- Digitonin; Thermo Fisher Scientific, Cat. no. BN2006
- L-Glutamic acid; Sigma-Aldrich Cat. no. G1251-1KG
- L-(-)-Malic Acid; Sigma-Aldrich Cat. no. M7397-25G
- KCl buffer:
  KCl; Sigma-Aldrich, Cat. no. 793590-1KG
  $K_2HPO_4$; Sigma-Aldrich, Cat. no. P3786-1KG
  $MgCl_2$; Sigma-Aldrich, Cat. no. M8266-1KG (not $MgCl_2$ hexahydrate?)
  Hepes; Sigma-Aldrich, Cat. no. H3375-1KG
- EGTA; Sigma-Aldrich Cat. no. E3889
- Carbonyl cyanide 3-chlorophenylhydrazone; Sigma-Aldrich Cat. no. C2759-250MG
- TMRM reagent; Thermo Fisher Scientific Cat. no. I34361
- Black 96-well plates; Greiner Bio-One Cat. no. 655076

Protocol and reagents adapted from Kovacs-Bogdan et al (2014).
Tetramethyl rhodamine methyl ester (TMRM) was used to assess the membrane potential of permeabilized cells. 1 million HEK293T cells were spun down at 800$g$ for 3 min in 1.5 ml microcentrifuge tubes, washed with 1 ml of PBS, and spun down again for 1 min at 800$g$. PBS was aspirated and cells were permeabilized in 150 $\mu$l KCl buffer (125 mM KCl, 2 mM $K_2HPO_4$, 1 mM $MgCl_2$, and 20 mM Hepes at pH 7.2, 0.005% digitonin) that was supplemented with 500 nM TMRM and 5 mM glutamate/malate from a 500 mM G/M stock solution that was filtered and stored at −20°C. Cell suspension was transferred to a black-bottom 96-well plate. Two readings of each sample were taken using a BioTek Synergy H1 microplate reader at room temperature. For each, a 540-nm excitation and 590-nm emission were recorded. The first reading was taken after cells' suspension in permeabilization buffer, to establish a baseline. Membrane potential was dissipated with the addition of 1 $\mu$M CCCP, and a second reading was taken after a 3-min incubation period. Each cell line was tested three times on the same day, for a total of six readings: three readings before the addition of CCCP (first readings), and three readings after the addition of CCCP (second readings). For the purposes of data analysis, the mean of the three "first" readings was calculated, as was the mean of the three "second" readings for each cell line. The error bars report the SD of these readings.

## Cloning

cDNA encoding for the chimeric proteins were generated by gene synthesis, cloned into pLYS1 (#19319; Addgene), or pLYS5 (#50054; Addgene) plasmids using NheI/EcoRI restriction sites.

MCU without any cysteines was generated by gene synthesis, and cysteine coding nucleotides at the desired locations were introduced by mutagenesis.

The sequences of all genes used in this study were verified by sequencing using CMV forward primer and custom designed reverse primer (TCTCGCACATTCTTCACGTC).

## HEK293T EMRE knockout cell line production

- eSpCas9(1.1) plasmid; Addgene plasmid #71814
- QIAquick PCR Purification Kit; QIAGEN, Cat. no. 28106
- dNTP set, PCR grade; QIAGEN, Cat. no. 201913
- QIAprep Spin Miniprep Kit; QIAGEN, Cat. no. 27106

- Q5 High Fidality DNA polymerase; QIAGEN Cat. no. M0491S
- HsEMRE gRNA: GCCGGAGCCTGGTACCGTCG

MCU KO cell line was described before (Sancak et al, 2013). EMRE gRNA was cloned into a gRNA expression plasmid. 600,000 cells growing on six-well plates were transfected with 250 ng of gRNA expression plasmid and 1 $\mu$g of eSpCas9(1.1) plasmid. 2 d later, cells were diluted at 1 cell/well and plated on 96-well plates to obtain single cell clones. EMRE KO cell clones were verified by Western blotting, functional assays, and by sequencing.

## Lentivirus production and infection

- X-treme(GENE) 9 DNA Transfection Reagent; Sigma-Aldrich, Cat. no. 6365779001
- psPax2; Addgene Cat. no. 12260
- VSV-G; Addgene, Cat. no. 8454
- Puromycin dihydrochloride; VWR, Cat. no. 62111-170
- Hygromycin B Solution; VWR, Cat. no. 45000-806
- Hexadimethrine bromide/polybrene; Sigma-Aldrich Cat. no. H9268-10G
- DMEM; Thermo Fisher Scientific, Cat. no. 11-965-118
- Filter; VWR, Cat. no. 28145-505
- Syringe; VWR, Cat. no. 28200-042

### Lentivirus production

1 million HEK293T cells were plated in 6-cm plates in 5 ml of media. 1 d later, the cells were transfected with viral mix. To prepare the viral mix, 100 ng VSV-G, 900 ng psPax2, 1 $\mu$g viral plasmid, dH$_2$O to 10 $\mu$l, and 150 $\mu$l DMEM mixed with 6 $\mu$l X-treme(GENE) and incubated for 30 min at room temperature before being added to the cells and mixed. 2 d later, the medium—which now contained the virus of interest—was filtered through a 0.45 $\mu$m sterile filter attached to a syringe. The virus was stored at −80°C until use.

### Lentivirus infection

250K HEK293T cells were plated in a six-well dish containing 2 ml media. The following day, 200 $\mu$l of the virus-containing media and 2 $\mu$l of polybrene from an 8-mg/ml stock solution was added to a final concentration of 8 $\mu$g/ml. Polybrene stock solution was prepared in water, filter sterilized and stored at −20°C for long term storage and at 4°C for short term storage. 2 d later, the cells were split and transferred to 10 cm tissue culture plates and selected. Chimera cell lines were selected using 1 $\mu$g/ml puromycin from a 1 mg/ml puromycin stock solution that was prepared in water, filter-sterilized, and stored at −20°C for long-term storage and at 4°C for short-term storage. Copper-mediated cysteine cross-linking cell lines infected with cysteine point mutations in MCU were selected using 100 $\mu$g/ml from a 50 mg/ml hygromycin stock solution that was prepared in water, filter sterilized, and stored at 4°C.

### Transient transfection of functional chimeras and EMRE

- X-treme(GENE) 9 DNA Transfection Reagent; Sigma-Aldrich, Cat. no. 6365779001
- DMEM; Thermo Fisher Scientific, Cat. no. 11-965-118

Two million cells of each functional chimera were plated in seven 10-cm plates. The next day, the cells were transiently transfected with functional chimeras and wild-type EMRE. To prepare the transient transfection mix, 1 μg chimera plasmid, 2 μg EMRE plasmid, dH$_2$O to 10 μl, and 150 μl DMEM mixed with 6 μl X-treme(GENE) and incubated for 30 min at room temperature before being added to the cells and mixed. 2 d after transient transfection, the cells were DSP cross-linked as described below and lysed as described above. A small fraction of the lysate was used to prepare samples at 1 μg/μl of protein concentration in reducing sample buffer for Western blot analysis of lysates. Lysate was then immunoprecipitated as described below. After immunoprecipitation, all samples were prepared in reducing conditions, electrophoresed, and Western blotted normally.

### Calcium uptake in permeabilized HEK-293T cells

• Digitonin; Thermo Fisher Scientific, Cat. no. BN2006
• Oregon Green 488 Bapta-6F; Invitrogen, Cat. no. O23990
• L-Glutamic acid; Sigma-Aldrich Cat. no. G1251-1KG
• L-(-)-Malic Acid; Sigma-Aldrich Cat. no. M7397-25G
• KCl buffer:
  KCl; Sigma-Aldrich, Cat. no. 793590-1KG
  K$_2$HPO$_4$; Sigma-Aldrich, Cat. no. P3786-1KG
  MgCl$_2$; Sigma-Aldrich, Cat. no. M8266-1KG (not MgCl$_2$ hexahydrate?)
  Hepes; Sigma-Aldrich, Cat. no. H3375-1KG
• EGTA; Sigma-Aldrich Cat. no. E3889

Protocol and reagents adapted from Sancak et al (2013). HEK-293T cells grown in 10-cm tissue culture plates were trypsinized and resuspended in 10 ml of prepared media. 1 million HEK-293T cells transferred to microcentrifuge tubes and spun down for 3 min at 800$g$ at room temperature to pellet cells. Cells were washed with PBS once and resuspended in KCl buffer (125 mM KCl, 2 mM K$_2$HPO$_4$, 1 mM MgCl$_2$, 20 mM Hepes, pH 7.2), supplemented with 5 mM glutamate/malate from a 500 mM G/M stock solution that was filtered and stored at –20°C, 0.005% digitonin, and 1 μM Oregon Green Bapta 6F.

For Fig 1, fluorescence was monitored every 0.2 s at room temperature using a Perkin-Elmer Envision plate reader before and after injection of 50 μM CaCl$_2$ using FITC filter sets (485 excitation and 535 emission). Calcium uptake rates were calculated using the linear fit of uptake curves between 20 and 30 s.

For all other figures, fluorescence was monitored for 78 s every 2 s at room temperature (~25°C) using a BioTek Synergy H1 microplate reader before and after injection of 50 μM CaCl$_2$ from a 500-μM stock prepared in dH$_2$O. Fluorescence was recorded using a fluorescent green filter set to 485/20 excitation, 528/20 emission. Calcium uptake rates were calculated using the linear fit of uptake curves between 20 and 30 s after calcium injection. A maximum of eight samples were assayed together, including one wild-type HEK293T control per assay run. To calculate calcium uptake rates relative to wild type, the wild-type rate for each sample set was set at 100%. Calculating $\frac{Experimental\ calcium\ uptake\ rate}{Wild\ type\ calcium\ uptake\ rate}$ yielded the relative calcium uptake rate. For each figure, each proportional experimental calcium uptake rate is plotted relative to its corresponding wild type calcium uptake rate.

### Crude mitochondria preparation

• 27.5-gauge needle; VWR, Cat. no. BD305109
• 1 ml syringe; BD Biosciences, Cat. no. 309659
• PBS; Thermo Fisher Scientific, Cat. no. 20012050

Cells were grown to confluency in 10-cm tissue culture plates. Cell culture medium was aspirated, and the culture plates were rinsed with 4°C PBS. PBS was aspirated and cells were harvested in fresh PBS. The cells were passed in and out of a 1-ml syringe through a 27.5-gauge needle 12 times. The disrupted mixture was then spun at 800$g$ for 5 min, 4°C, to pellet nuclei and intact cells. The supernatant was then spun at 8,000$g$ for 5 min, 4°C, to pellet mitochondria. The resulting supernatant was then aspirated. The pellet was used for downstream applications.

### Immunoprecipitation

• Anti-FLAG M2 Affinity Gel, Sigma-Aldrich Cat. no. A2220-5ML

Cells were grown to confluence in 10-cm tissue culture plates. Cell culture media was aspirated, and the culture plates were rinsed with 4°C PBS. PBS was aspirated and cells were harvested and lysed on ice in 700-ml lysis buffer (supplemented with 1% Triton and protease inhibitors). Cells were spun for 10 min at 4°C, maximum speed. The protein concentration was determined using a Bradford assay. A small fraction of the lysate was used to prepare samples at 1 μg/μl of protein concentration in reducing sample buffer for Western blot analysis of lysates. Between 1 and 3 mg of protein was used for IP experiments. Lysates were incubated with 10 μl of anti-FLAG M2 affinity gel beads (from the gel's 1:1 bead:glycerol slurry). Beads were washed three times beforehand with 1 ml 1% Triton lysis buffer. Volumes of all IP samples were standardized. IP samples were rocked on a nutator for 2–4 h at 4°C. Following aspiration of unbound lysate, samples were washed with 1 ml 1% Triton lysis buffer three times before mixing with sample buffer (reducing or non-reducing, depending on application) and boiled for 5 min at 95°C.

### DSP crosslinking

• DSP; Thermo Fisher Scientific, Cat. no. 22585
• DMSO; Sigma-Aldrich, Cat. no. D8418-500ML
• Trizma base; Sigma-Aldrich Cat. no. RDD008

DSP was dissolved in DMSO to a final concentration of 250 mg/ml to make a 250× stock solution for the in-cell cross-linking assay. 40 μl of DSP solution was then added to confluent cells growing in 10 cm plates in 10 ml media. After swirling DSP to mix it into the media, cells were incubated at room temperature for 3 min. The reaction was quenched by adding 1 ml 1M Tris, pH 8.0, to the plates and swirling again. The medium was aspirated, plates were washed with 4°C PBS, and cells were lysed using 1% Triton lysis buffer (50 mM Hepes KOH, pH 7.4, 150 mM NaCl, 5 mM EDTA, and 1% Triton). MCU was immunoprecipitated as described above.

### Copper-mediated cysteine cross-linking

• Anti-FLAG M2 Affinity Gel; Sigma-Aldrich Cat. no. A2220-5ML
• Phenanthroline; Sigma-Aldrich, Cat. no. 131377-5G
• CuSO$_4$; Sigma-Aldrich, Cat. no. 6365779001
• EDTA; Sigma-Aldrich, Cat. no. 607-429-00-8

Cu(II)-(1,10-phenanthroline)$_3$ was prepared in PBS by combining 100 $\mu$M CuSO$_4$ + 300 $\mu$M phenanthroline (8 ml PBS + 0.8 $\mu$l 1M CuSO$_4$ + 80 $\mu$l 30 mM phenanthroline). Crude mitochondria were resuspended in 250 $\mu$l copper phenanthroline solution and incubated for 20 min at room temperature. The cross-linking reaction was stopped by the addition of 10 mM EDTA from a 500 mM stock prepared in dH$_2$O, pH 8. Mitochondria were spun and pelleted at 8,000$g$ for 5 min, and supernatant was aspirated with a needle and discarded. Mitochondria were then lysed with 200 $\mu$l 4°C lysis buffer (50 mM Hepes KOH, pH 7.4, 150 mM NaCl, 5 mM EDTA, and 1% Triton, protease inhibitors). A small fraction of the lysate was used to prepare samples at 1 $\mu$g/$\mu$l protein concentration in reducing sample buffer for Western blot analysis of lysates. 150–200 $\mu$g lysate protein was affinity-purified as described above under "Immunoprecipitation." After immunoprecipitation and washing, the beads were boiled with 20 $\mu$l non-reducing 2.5× sample buffer. 18 $\mu$l of the resulting samples were electrophoresed under non-reducing conditions and Western blotted to detect EMRE; 2 $\mu$l of each sample was separated, mixed with 18 $\mu$l 2.5× sample buffer, electrophoresed under non-reducing conditions, and Western blotted to detect MCU-FLAG.

### Mitoplast preparation and PEG5K-Maleimide conjugation

- PEG5K-maleimide; Sigma-Aldrich, Cat. no. 363187, 10 mM stock solution was prepared in DMSO

Crude mitochondria were prepared in PBS from one confluent 15-cm plate of cells using a 27.5-g needle as described. To prepare mitoplasts, 450 $\mu$g of crude mitochondrial prep was resuspended in 375 $\mu$l of cold dH$_2$O and incubated on ice for 10 min to swell. After 10 min, 125 $\mu$l of 4× respiration buffer was added (4× respiration buffer: 548 mM KCl, 40 mM Hepes, pH 7.2, 10 mM MgCl$_2$), and the tubes were vortexed briefly. The samples were spun down for 3 min at 4°C at 800$g$, resuspended in 1× respiration buffer to 3 $\mu$g/$\mu$l. 10 $\mu$l of this mitoplast preparation was incubated with 1 mM PEG5K-maleimide dissolved in DMSO in 30 $\mu$l final volume for 30 min at RT in 1× respiration buffer, in the absence or presence of 1 $\mu$l of 10% Triton X-100 dissolved in dH$_2$O. Reaction was stopped by addition of 6 $\mu$l of 5× SDS sample buffer. The lysates were subjected to SDS–PAGE and Western blotting.

### PK treatment of crude mitochondrial preparations

- PK; Sigma-Aldrich, Cat. no. P2308, prepared in IBc buffer as a 1,000× stock solution
- PMSF; Sigma-Aldrich, Cat. no. P7626, prepared in ethanol as a 100× stock solution.

20 $\mu$g of crude mitochondria isolated from cultured cells as described were treated with 100 $\mu$g/ml of PK in the presence of increasing concentrations of digitonin in 30 ml of final volume in IBc buffer for 15 min at room temperature, 7 mM of PMSF was added inactivate PK for 5 min. 5 ml of 5× SDS sample buffer was added, samples were boiled, and 5–10 $\mu$l was loaded on a Tris–glycine gel for Western blotting.

### BN-PAGE

- Running Buffers; Invitrogen NativePAGE Novex Bis-Tris Gel System, Cat. no. BN1001BOX, BN1002BOX, and BN1004BOX

- Protein Standard; Invitrogen NativeMARK Unstained Protein Standard, Cat. no. LC0725
- 3–12% Bis-Tris Gel; NativePAGE 3–12% Bis-Tris Gel, Cat. no. BN1003BOX

Protocol and reagents were adapted from Sancak et al (2013). Gel electrophoresis running buffers were prepared according to the manufacturer's protocol for the Invitrogen NativePAGE Novex Bis-Tris Gel System. Running buffers were cooled to 4°C before use, and electrophoresis was performed at 4°C. Invitrogen NativeMark Unstained Protein Standard was used to estimate molecular weight. Gels were run at 40 V for 30 min. Voltage was then increased to 100 V for 1 h, and subsequently to 250 V for 90 min. When the dye front had traveled through ~1/3 of the gel, electrophoresis was paused, and the Dark Blue Cathode Buffer was replaced with Light Blue Cathode Buffer, as per the manufacturer's protocol.

### Blue native PAGE transfer

- Transfer apparatus; Bio-Rad Trans-Blot SD cell
- Blotting paper; Bio-Rad extra thick blot paper, Cat. no. 1703965
- Acetic acid; Thermo Fisher Scientific, Cat. no. A38C-212
- Ethanol: 200 proof (100%); Thermo Fisher Scientific Cat. no. 04-355-450

Protocol and reagents adapted from Sancak et al (2013). After electrophoresis was complete, the gels were transferred to Bio-Rad Mini-size 0.22 $\mu$m PVDF membranes in Invitrogen Novex Tris–glycine transfer buffer at 0.18 A for 20 min, using a Bio-Rad TransBlot SD Semi-Dry Transfer Cell and extra thick blotting paper. Membranes were incubated in 8% acetic acid while shaking for 15 min to fix the proteins. The membranes were rinsed with dH$_2$O for 5 min, and then air-dried. Once dry, the membranes were rehydrated with ethanol. The membranes were then blocked with 5% milk in TBST (wt/vol) and immunoblotted using FLAG antibody as described above. Finally, the same membranes were probed for ATP5A (a mouse antibody) as described above, as a loading control.

### Data reporting and statistical analysis

No statistical methods were used to predetermine sample size. The experiments were not randomized. The investigators were not blinded to allocation during experiments and outcome assessment. All quantitative experiments are presented as means ± SD of at least three independent biological experiments (as indicated).

## Supplementary Information

## Acknowledgements

We thank Jason McCoy and all members of the Mootha and Sancak laboratories for critical reading of the manuscript. We also thank David M

Shechner for help with Pymol, analysis of published MCU structures, and with gRNA expression advice. MJS MacEwen was supported by T32GM007750. This work was supported by grants from the National Institutes of Health (R01HL130143, R01AR071942) to VK Mootha. VK Mootha is an Investigator of the Howard Hughes Medical Institute.

## Author Contributions

MJS MacEwen: conceptualization, data curation, formal analysis, validation, visualization, methodology, and writing—original draft, review, and editing.

AL Markhard: data curation, investigation, and methodology.

M Bozbeyoglu: data curation and investigation.

F Bradford: data curation and investigation.

O Goldberger: resources, data curation, and investigation.

VK Mootha: conceptualization, funding acquisition, investigation, and writing—original draft, review, and editing.

Y Sancak: conceptualization, data curation, formal analysis, supervision, funding acquisition, validation, investigation, visualization, methodology, and writing—original draft, review, and editing.

## Conflict of Interest Statement

VK Mootha is a paid advisor to Janssen Pharmaceuticals and 5am Ventures.

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
