## [Reviewer comments · Life Science Alliance]

Life Science Alliance

Evolutionary divergence reveals the molecular basis of EMRE dependence of the human MCU

Melissa MacEwen, Andrew Markhard, Mert Bozbeyoglu, Forrest Bradford, and Olga Goldberger, Vamsi Mootha and Yasemin Sancak

DOI: <https://doi.org/10.26508/lsa.202000718>

Corresponding author(s): Yasemin Sancak, University of Washington and Vamsi Mootha

Review Timeline:

Submission Date:	2020-03-28
Editorial Decision:	2020-03-30
Revision Received:	2020-07-13
Editorial Decision:	2020-07-28
Revision Received:	2020-07-28
Accepted:	2020-07-29

Transaction Report:

Please note that the manuscript was previously reviewed at another journal and the reports were taken into account in the decision-making process at Life Science Alliance.

Referee #1 Review

Report for Author:

This manuscript explores the interaction of EMRE with MCU to attempt to understand why animal MCU requires EMRE as an essential component of the channel complex for its function. The authors employ a chimera approach of using EMRE-dependent and independent MCUs to hone in on a region of MCU that is required for EMRE regulation of MCU activity. This stretch of residues is essentially the same as that previously described by Wang et al 2019. They also perform crosslinking studies to identify regions of MCU-EMRE interactions, but confirming what was revealed in the cryo-EM structure of Wang et al. They develop a model of channel gating involving the region that they identified, borrowing from Wang et al that this is an inner gate. They speculate that MICU proteins regulate an inner gate but not an outer gate, without evidence.

This manuscript obviously benefitted from the Wang et al structure of human MCU in complex with EMRE, but it instead presents the studies as if they were conceived and conducted without these insights, and then comment that they are consistent with the structure. The manuscript should be re-written in a way that acknowledges the structure as providing the rationale for many of the specific mutations created, and indeed the for the ideas regarding the JML loop and its purported function and the model regarding an inner gate.

1. I don't really understand the title. Why is "evolutionary divergence" mentioned? Furthermore, the title is misleading as it implies that the regulation of MCU activity by MICU proteins is explored and explained. However, only one experiment with over-expression of a mutant MICU2 was performed, and the results (below) do not allow for unambiguous conclusions regarding the interactions of EMRE with MCU on the mechanism of regulation by MICU proteins.
2. The Ca²⁺ uptake curves shown in Fig 1A and Supp Fig 5 are unusual in demonstrating 2 phases..an initial slow phase that lasts for fully 20 sec, and then a more rapid one over the next 20 sec. This is in contrast to reports from other labs that have observed nearly instantaneous rapid uptake that necessitates determining initial Ca²⁺ uptake rates immediately after Ca²⁺ bolus injection. This raises some concern about the measurement system used here and the meaning of the measured uptake rates. This is particularly important because the rates are compared between different cell lines, i.e. they have been interpreted quantitatively.
3. Uptake rates are determined in part, because of thermodynamics and regulation, by the bolus of free Ca injected, but we don't have that information regarding the free Ca²⁺ concentration at the beginning of the uptake in any of the studies.
4. Figure legend 1(F) and 1(G) are missing.
5. Membrane topology of EMRE was previously established. I understand why they might want to repeat those experiments here, but a rationale for doing these experiments should be presented right at the beginning so the reader knows that the authors want to validate previously reported data.
6. p. 5, line 5 from bottom. In addition to refs 38 and 40, Vais et al Cell Reports demonstrated that charge neutralization of EMRE C-terminus did not inhibit mitochondrial Ca²⁺ uptake or MCU Ca²⁺ currents. It should also be cited.
7. P7. The authors use chimeric MCU proteins to hone in on a 10-residue stretch that when transplanted into human MCU from Dictyostelium MCU confers EMRE-independent MCU activity. This domain is here termed the EMRE dependence domain (EDD). This is the same stretch or amino acids identified by Wang et al Cell 2019, which was termed by them as the JML. I wonder if it is wise at this time to use different terminology for the same region of MCU. Please justify.
8. Furthermore, Wang et al already previously demonstrated the requirement for this stretch of residues for EMRE dependence/independence. What is novel in this study?
9. P10, first sentence. The authors state that their functional data highlight the importance of MCU TM domains in the EMRE-dependence of MCU, but they only demonstrated the importance of the JML region.
10. P11. The authors claim to have established for the first time that MCU TM2 interacts with

EMRE. However, Wang et al in their structure manuscript previously described an interaction of EMRE with TM2.

11. Regarding the DSP crosslinking experiments:

a) Chimera 9 does not require EMRE for function, yet it is cross-linked to EMRE. Chimera 1 requires EMRE and crosslinks as well as chimera 9. Chimera 5 requires EMRE, but it doesn't crosslink. This is a non-coherent set of data. How do the authors interpret these seemingly paradoxical results?

12. Regarding the copper crosslinking experiments...

a) What was the rationale for using MCU-KO cells rather than double MCU/EMRE KO cells? The presence of endogenous EMRE in these experiments in cells into which cys-EMRE was transfected complicates interpretations about whether these mutant EMRE are still functional.

b) Fig 1B demonstrates that EMRE and MCU can be co-immunoprecipitated without crosslinking. Are the conditions more stringent here (Fig 3A,B; Fig 4A-C) so only cross-linked interactions are preserved? If so, why is un-cross-linked EMRE sometimes observed?

c) In transient transfections of EMRE, an unprocessed form is often the dominant species, revealed at doublets in western blots, whereas the western blots here show a single band. Is this the fully-processed form or not?

d) What was the rationale for choosing the specific EMRE and MCU residues for cys-mutagenesis? If it was based on the hMCU/EMRE structure, this should be stated.

e) What amino acids replaced the cys in the cys-less MCU?

f) In Fig 3C,D, please state what antibody was used for the IP blots.

g) In Fig 4C. The authors make conclusions about "flexibility in this region of the complex". What do the authors mean? How does cross-linking to more than one residue provide information about protein dynamics or stability? There is a conclusion on page 13 that EDD is "flexible", but what is the evidence? And that it blocks exit from the pore...what is the evidence? There is too much speculation without experimental evidence.

h) P13. The authors suggest that they have demonstrated that the "N-terminus of EMRE is critical for the EMRE-EDD interaction", implying, based on the earlier part of the sentence, that it is critical for the EMRE-MCU interaction, but I don't think that this has been shown here. Furthermore, they state in the Discussion that there are likely additional extended interactions between MCU and EMRE

13. Fig 4E-G are interesting. Can the authors explain the rationale for choosing *C. elegans*? What is the sequence conservation/divergence in these regions that might account for these results?

14. I think that reference to the tetrameric JML in the Wang et al structure as "the Ca²⁺ exit site" of MCU is premature, even though Wang et al made the same claim. There is no evidence that Ca²⁺ needs to traverse this portal to enter the matrix, or that it even does. The structure is suggestive, but without functional evidence it is still an hypothesis. Perhaps the use of a word such as "putative" might be prudent. Furthermore, the authors state that chimera 9 has a constitutively open "Ca²⁺ gate distal to the pore", but it has not been shown that it is indeed a gate, nor that it is either constitutively closed or open under different conditions here. I think caution is warranted until there are experimental data.

15. The authors interpret lack of apparent inhibition by mutant MICU2 of MCU activity as evidence that MICU1/2 modulates the "gate distal to the pore", but lack of inhibition does not imply that MICU1/2 does not regulate Ca²⁺ access into the pore, i.e. there is no evidence to support this statement. Another way of thinking about the data is that the mutant chimera 9 channel modulates the channel structure in a way that impinges on MICU1/2 regulation. Additional

experimentation would be necessary to dissect these alternatives.

16. P17, line 7, please insert a reference citation.

17. Discussion paragraph 2. The human MCU/EMRE structure isn't mentioned at all here, whereas it's the focus of this study. Clearly, the Wang et al structure informed many of the experiments in this manuscript and the authors should acknowledge this.

18. P17. The authors state that fungal MCU channels have lower conductance than animal MCU, but what is the evidence for this...it is not present in the references cited.

Referee #2 Review

Report for Author:

This manuscript reports analysis to identify the EMRE and MCU interacting domains based on the use of chimeras between human and dictyostelium MCU and partial cysteine scan. They also use one of the chimera to examine a new potential role of MICU1/2 in the regulation of the MCU pore.

Although the results are fairly clean, unfortunately, they add very little to the structure of MCU/EMRE published in May of last year. The structure provides a more detailed information and makes it quite difficult to support publication of this manuscript in this journal. The only additional and new findings are in Figure 5, however, they are preliminary as presented. The lack of effect of MICU2 can be for multiple reasons such as altered interaction of the chimera with MICU2 (lower affinity), altered interaction with MICU1 that is required for the effect of MICU2, altered Ca²⁺-dependence of Ca²⁺ uptake by the chimera, among others. More rigorous characterization of the chimera, including biophysical analysis of chimera pore, is needed to claim a role for MICUs on channel gating through interaction between MCU-EMRE.

Referee #3 Review

Report for Author:

Mitochondrial calcium uptake is critical for mitochondrial and cellular function. The uptake route relies on the channel MCU which interacts with numerous regulatory/accessory proteins. One of these is the protein EMRE which interestingly appears not to be co-conserved with MCU. Still EMRE is critical for MCU function in organisms containing EMRE. This implies that there might be features in (human) MCU that are important for its functional dependence on EMRE. In this study, the authors set out to identify these regions using chimeras of EMRE-dependent and EMRE-independent MCU variants. They thereby identify a region in MCU critical for EMRE interaction and characterize it further biochemically.

This is a very nice piece of high-quality biochemical work. My major concern affects the conceptual novelty of the findings: the interaction site of EMRE and MCU has already been previously identified in MCU (Wang et al) - although in a slightly different amino acid window. The current study expands on this but does not significantly address the in my opinion critical and really novel question of the evolutionary and regulatory significance of EMRE.

Further points:

Fig. 2: complementation studies with MCU chimeras were performed in either MCU or EMRE single KO cells. Why did the authors not use MCU-EMRE double knockout cells to exclude e.g. interference of endogenous MCU with chimeras in EMRE KO cells?

Fig. 4: Cysteine crosslinking data (interaction between far-away residues) - the authors interpreted their findings as "flexible nature" of the interaction. How can they exclude artefacts from the experiment? What is the functional relevance of this flexible interaction? Is the interaction dynamic?

Figure 5: MICU1 binding to MCU does not result in closing of chimera 9. This implies that EMRE is hierarchically more important for the control of opening and closing. However, compared to MICU1, its binding to MCU appears not to be dynamically regulated? Can the authors further elucidate on this and hypothesize how the non-dynamic binding of EMRE helps in regulating MCU activity.

p.3, 2nd paragraph: "Intermembrane" should read "intermembrane space"

March 30, 2020

RE: Life Science Alliance Manuscript #LSA-2020-00718-T

Dr. Yasemin Sancak
University of Washington
Department of Pharmacology
1959 NE Pacific Street
Rm K536B
Seattle, Washington 98195-7750

Dear Dr. Sancak,

Thank you for transferring your manuscript entitled "Evolutionary divergence reveals the molecular basis of EMRE dependence of the human mitochondrial calcium uniporter and its regulation by MICU1/2" to Life Science Alliance. Your manuscript was reviewed at another journal before, and the editors transferred those reports to us with your permission.

The reviewers who evaluated your study elsewhere noted that your data are of high quality and robustness. However, they would have expected a further reaching conceptual advance. This does not preclude publication here, and I would thus like to invite you to submit a revised version of your manuscript to us. Please provide a point-by-point response to all reviewer comments and accordingly change the data representation and discussion, also leaving room for alternative explanations where needed given the input received. No further experiments are needed, so we assume that the revision is feasible despite the current SARS-CoV-2 pandemic and the associated difficulties many labs experience.

Additionally, please pay attention to the following:

- please add the number of replicates to the figure legends where currently missing
- please upload all figure, including S figures, as individual files
- please provide your manuscript file in docx format- please include the figure legends, including the supplementary figure legends, in the main manuscript docx file
- please add the info to Fig 1F and G to the legend
- please list 10 authors et al in your reference list
- please fill in all mandatory fields in our submission system and make sure that the author order in our system matches the one in your manuscript file
- please make sure that all corresponding authors link their ORCID iD to their profile in our submission system, they should have received an email with instructions on how to do so

You will be guided to complete the submission of your revised manuscript and to fill in all necessary

information. Please get in touch in case you do not know or remember your login name.

A. FINAL FILES:

B. MANUSCRIPT ORGANIZATION AND FORMATTING:

Thank you for your attention to these final processing requirements.

Sincerely,

We thank the reviewers for careful reading and critique of our manuscript. We addressed all of their concerns below.

Referee #1:

This manuscript explores the interaction of EMRE with MCU to attempt to understand why animal MCU requires EMRE as an essential component of the channel complex for its function. The authors employ a chimera approach of using EMRE-dependent and independent MCUs to hone in on a region of MCU that is required for EMRE regulation of MCU activity. This stretch of residues is essentially the same as that previous described by Wang et al 2019. They also perform crosslinking studies to identify regions of MCU-EMRE interactions, but confirming what was revealed in the cryo-EM structure of Wang et al. They develop a model of channel gating involving the region that they identified, borrowing from Wang et al that this is an inner gate. They speculate that MICU proteins regulate an inner gate but not an outer gate, without evidence.

This manuscript obviously benefitted from the Wang et al structure of human MCU in complex with EMRE, but it instead presents the studies as if they were conceived and conducted without these insights, and then comment that they are consistent with the structure. The manuscript should be re-written in a way that acknowledges the structure as providing the rationale for many of the specific mutations created, and indeed the for the ideas regarding the JML loop and its purported function and the model regarding an inner gate.

Response: We disagree with this comment that “this manuscript obviously benefitted from the Wang et al structure of human MCU.” This is not at all true -- the reviewer is obviously not aware of the bioRxiv version of our manuscript (published on May 14 2019), only 5 days after Wang et al's MCU/EMRE structure was published online on May 9 2019. In stark contrast to what Reviewer #1 is suggesting, all of our experiments that involved MCU-EMRE interaction and function were conceived, performed and interpreted in the absence of the structure published by Wang et al. These data and their implications for EMRE-dependence, and functional data on chimera experiments, have been openly shared with the scientific community at scientific meetings, seminars in 2016-2017, including many authors of recent structural biology papers.

It is notable that our functional mutagenesis and crosslinking data is largely concordant with what has been published in the structure, which is important for the field. We will note that the EDD domain we have identified shows some, but not complete, overlap with the JML, such that our functional studies here are consistent with the reported structure. We report eight crosslinking events, of which six seem plausible in the context of the structure. We acknowledge now in the revised manuscript discussion that two of these cannot be explained by the structure, and hence, future experiments will be required to reconcile them and establish their physiological relevance. They appear as robust as the other structure validated crosslinks. We acknowledge in the revised discussion that these represent new features of the channel complex we don't understand, though we can't exclude the possibility of an artifactual crosslinking event. Finally, there are numerous other experimental nuggets we have included that are important data points for the field.

In the revised version of the manuscript that has been transferred to LSA, we stressed that fact that our bioRxiv submission coincides with MCU-EMRE structure paper (Wang et al). Our manuscript is extremely valuable to the field since the functional mutagenesis and cross-linking experiments were performed in a manner not biased by the structure. Hence, it can be viewed as an independent set of experiments. Crucially the Wang et al paper utilized the same experimental expression system that our group first pioneered and proposed, the HEK-293T cell system for expressing and purifying the uniporter (Sancak *Science* 2013). We also explain that the structure shown by Wang et al does not provide the rationale for any of the specific mutations created: based on the structure, at least two the residues that we show to crosslink, should not crosslink. This could be due to multiple reasons: (a) structure shows only one confirmation of MCU/EMRE complex (b) resolution of the structure is low (uncertainty of side chain assignments are particularly high) in the MCU transmembrane 2 and EMRE interaction surface, or (c) as we acknowledge, there can be artifactual cross-linking.

1. I don't really understand the title. Why is "evolutionary divergence" mentioned? Furthermore, the title is misleading as it implies that the regulation of MCU activity by MICU proteins is explored and explained. However, only one experiment

with over-expression of a mutant MICU2 was performed, and the results (below) do not allow for unambiguous conclusions regarding the interactions of EMRE with MCU on the mechanism of regulation by MICU proteins.

Response: We used “evolutionary divergence” in the title to highlight the fact that we used systematically investigated MCU from human (requiring EMRE) and from Dicty (not requiring EMRE), to understand EMRE dependence. This divergence enabled us to map a region in MCU protein that reveals its EMRE-dependence. In the revision we have removed the final experiment with MICU2 (which was deemed too preliminary) and have revised the title so it is congruent with our original bioRxiv paper. However, this does retain the “evolutionary divergence” which is crucial as indicated above.

2. The Ca²⁺ uptake curves shown in Fig 1A and Supp Fig 5 are unusual is demonstrating 2 phases..an initial slow phase that lasts for fully 20 sec, and then a more rapid one over the next 20 sec. This is in contrast to reports from other labs that have observed nearly instantaneous rapid uptake that necessitates determining initial Ca²⁺ uptake rates immediately after Ca²⁺ bolus injection. This raises some concern about the measurement system used here and the meaning of the measured uptake rates. This is particularly important because the rates are compared between different cell lines, i.e. they have been interpreted quantitatively.

Response: Our calcium uptake traces are linear during 20-50 sec after calcium injection. In the manuscript, we calculate calcium uptake rates using data from 20-30 second after calcium injection, and the relative calcium uptake rates are comparable whether we use 20-30 or 20-50 seconds after injection (please see the figure below).

We do not see a rapid, fast uptake due to two reasons:

- We use a plate reader for calcium uptake experiments and there's ~3-5 sec delay between calcium injection, shaking the plate and starting the measurement again.
- We use Oregon Green Bapta 6F in our experiments, which has ~3 μ M affinity for calcium. We inject ~35 μ M of free calcium. The dye remains “saturated with calcium” and the signal intensity does not change dramatically until free extramitochondrial calcium concentration reaches lower levels, giving the impression that there's a slow uptake at the beginning. Like every experiment, this experimental readout is a function of the reagents used. If we had used a calcium dye with high affinity for calcium (for example Fluo4 with a K_d of ~350nM for calcium), this curve would look much different, almost as if there's no calcium uptake for the first half of the experiment.

Our curves may look different from some other data, for example this data pasted below from PMID: 23101639 that also injects 50 μ M of calcium to reach ~30 μ M of free extramitochondrial calcium due to several reasons, 3 of which are listed below:

-FuraFF with a K_d of $\sim 6\mu\text{M}$ for calcium is used, as opposed to Oregon Green Bapta 6F, with a K_d of $3\mu\text{M}$ for calcium
 -The experiment is carried out in a well stirred cuvette, which allows for better mixing, oxygenation of the sample and simultaneous monitoring of signal
 -6 million cells are used here, we use 1 million cells in our experiments, which will lead to approximately 6 times faster reduction of extramitochondrial calcium levels.

In all our experiments, we report calcium uptake rates relative to wild type HEK293T cells that were processed together with the experimental samples. When we compare calcium uptake rates of wild type cells across time (see table below), we do not see a significant variation in rates (a $\sim 10\%$ variation between samples processed on different days). This enables comparison of calcium uptake rates between cell lines since they are reported as “relative to wild type”.

Date	Cell type	Calcium uptake rate relative to Dec 15
15-Dec	WT 293T	1
17-Dec	WT 293T	1.056000385
18-Oct	WT 293T	1.11995221

Consequently, our calcium measurements are robust and report uniporter function under the same experimental conditions for a set of chimeric proteins and mutants. We published several papers using this assay and all of our findings have been reproducible. In addition, several publications use similar relative calcium uptake rates, including Wang et al.

3. Uptake rates are determined in part, because of thermodynamics and regulation, by the bolus of free Ca injected, but we don't have that information regarding the free Ca^{2+} concentration at the beginning of the uptake in any of the studies.

Response: As mentioned above, we report calcium uptake rates relative to a WT control in all of the experiments and this relative rate is a unitless measurement. Free calcium concentration after $50\mu\text{M}$ calcium injection is $\sim 35\mu\text{M}$, however, this number is irrelevant for our rate comparison.

4. Figure legend 1(F) and 1(G) are missing.

Response: Thank you for catching this mistake. They have been added.

5. Membrane topology of EMRE was previously established. I understand why they might want to repeat those experiments here, but a rationale for doing these experiments should be presented right at the beginning so the reader knows that the authors want to validate previously reported data.

Response: We added the following sentence that indicates the rationale for the experiment in the main text.

6. p. 5, line 5 from bottom. In addition to refs 38 and 40, Vais et al Cell Reports demonstrated that charge neutralization of EMRE C-terminus did not inhibit mitochondrial Ca^{2+} uptake or MCU Ca^{2+} currents. It should also be cited.

Response: We apologize for unintentional omission of Vais et al. This paper is now been added as reference.

7. P7. The authors use chimeric MCU proteins to hone in on a 10-residue stretch that when transplanted into human MCU from Dictyostelium MCU confers EMRE-independent MCU activity. This domain is here termed the EMRE dependence domain (EDD). This is the same stretch or amino acids identified by Wang et al Cell 2019, which was termed by them as the JML. I wonder if it is wise at this time to use different terminology for the same region of MCU. Please justify.

Response: EDD only covers 3 of 6 amino acids of JML as shown by the alignment below. We show crosslinking between MCU and EMRE in a region of EDD that does not overlap with JML (E293 and D296). In

addition, our nomenclature stems from a functional readout and is unique to MCU. JML annotation is not unique to MCU, it's a structural region that many other proteins have, and does not have a function associated with it. We would like to keep EDD nomenclature because (1) it's not identical to JML (2) it's not a structural region but encompasses two structural domains (JML and adjacent coiled coil 2 of MCU) (3) communicates the function of the domain.

8. Furthermore, Wang et al already previously demonstrated the requirement for this stretch of residues for EMRE dependence/independence. What is novel in this study?

Response: Again this referee is not aware of the fact that the pre-print version of our paper was published 5 days after the publication of Wang et al. They are highly complementary to each other, ours is rich with functional mutagenesis, chimera analysis and physiology, while theirs is largely structures. Novel findings in our paper is summarized here:

(a) We identify EDD, which is overlapping but distinct from JML. Comparison of functional data between different chimeras (compare chimera 9 to chimeras 6 and 7) clearly show the importance of residues outside of JML for EMRE function.

(b) We show more extensive interactions between EMRE and MCU pore forming TM2 that were not detected in the structure. Wang et al claims that EMRE and MCU interact only at TM1 and around the JML in TM2 (around MCU amino acids 280- 300). We find robust and reproducible interaction between MCU and EMRE at amino acid 270, which is closer to the intermembrane space. We now discuss in the manuscript that these differences can be due to technical issues or due to the dynamic nature of the complex *in vivo*. These results also caution the community against taking the structure as the only and ultimate confirmation for MCU-EMRE complex; argues for the presence of other, functionally important MCU-EMRE interactions *in vivo*; and makes a strong case for additional functional experiments to understand how this interesting calcium channel functions at the molecular level *in vivo*.

(c) We provide data for the first time that interactions between EMRE N-terminal domain and MCU EDD are important for both binding and function of the channel (Figures 4F and 4G). Previous studies suggested that MCU TM1 and TM2 are the determinants, but our data clearly show that even though TM domains of MCU and EMRE are wild type sequences from the same species, incompatibility at EMRE's N-terminal domain and MCU's EDD lead to loss of binding between the two.

9. P10, first sentence. The authors state that their functional data highlight the importance of MCU TM domains in the EMRE-dependence of MCU, but they only demonstrated the importance of the JML region.

Response: We acknowledge this assertion was made only indirectly in our manuscript, and regret it was not presented more clearly. Comparison of DdMCU and Chimera 4 functional data demonstrate that the presence of DdMCU EDD alone does not ensure EMRE independence (see figure below). Rather, based on chimera 4's EMRE dependence, the transmembrane domains also play key roles in EMRE dependence of HsMCU. We now added the sentence below in the main text to make this conclusion clearer. "Chimera 4 showed reduced, but EMRE-dependent uniporter activity (Figure 2C), suggesting that the TM domains of HsMCU are involved in its EMRE-dependence."

10. P11. The authors claim to have established for the first time that MCU TM2 interacts with EMRE. However, Wang et al in their structure manuscript previously described an interaction of EMRE with TM2.

Response: We apologize we forgot to change this sentence in our manuscript submission. This claim was rightfully made when we first wrote the manuscript before the structure was published and was in the bioRxiv submission and was kept in this submission unintentionally. We now refer to Wang et al as well and add the following sentence in the main text: “Collectively, these findings confirm the interaction that was observed between MCU TM1 and EMRE previously (38), and establish that the pore-forming TM2 of MCU also interacts with EMRE, both within and outside the MCU-EMRE interaction domain defined by the MCU-EMRE structure (Wang et al)”

11. Regarding the DSP crosslinking experiments:

a) Chimera 9 does not require EMRE for function, yet it is cross-linked to EMRE. Chimera 1 requires EMRE and crosslinks as well as chimera 9. Chimera 5 requires EMRE, but it doesn't crosslink. This is a non-coherent set of data. How do the authors interpret these seemingly paradoxical results?

Response: Chimera 9 does not require EMRE for function, yet it is cross-linked to EMRE – Correct

Chimera 1 requires EMRE and crosslinks as well as chimera 9- Correct

Chimera 5 requires EMRE, but it doesn't crosslink- Incorrect. Chimera 5 does not require EMRE to function. It is EMRE-independent (Figure 2C). As a result, these results are not paradoxical.

We also would like to stress the fact that EMRE-binding and EMRE-dependence are independent from each other, and binding does not necessitate dependence in function (as in Chimera 9).

12. Regarding the copper crosslinking experiments...

a) What was the rationale for using MCU-KO cells rather than double MCU/EMRE KO cells? The presence of endogenous EMRE in these experiments in cells into which cys-EMRE was transfected complicates interpretations about whether these mutant EMRE are still functional.

Response: When we started these experiments, we did not have access to MCU/EMRE double KO cells. We agree with the reviewer that it would have been better to check function of every cys-EMRE residue expressed in EMRE KO cells. In unpublished results, we performed an alanine mutagenesis screen of EMRE amino acids that are in its transmembrane domain (EMRE aa 65-85). All of the alanine substitutions, except for amino acids 69 and 81, did not disrupt EMRE function. Although these are cysteine substitutions, we expect them to be functional similar to alanine mutations.

b) Fig 1B demonstrates that EMRE and MCU can be co-immunoprecipitated without crosslinking. Are the conditions more stringent here (Fig 3A, B; Fig 4A-C) so only cross-linked interactions are preserved? If so, why is un-cross-linked EMRE sometimes observed?

Response: We apologize for leaving this important point from materials and methods. In Figure 1B, 0.2% DDM is used. This milder detergent preserves MCU-EMRE interaction however recovery of the protein

complex from the membrane is not complete. In cysteine and DSP crosslinking experiments, we use 1% Triton X-100 during lysis to increase recovery of the complex from the membrane and increase yield. We clarified this in materials and methods section now.

c) *In transient transfections of EMRE, an unprocessed form is often the dominant species, revealed at doublets in western blots, whereas the western blots here show a single band. Is this the fully-processed form or not?*

Response: In our transient transfection experiment (Figure 3A), we do not observe an unprocessed EMRE band (see below uncropped western blots for this experiment). The presence of unprocessed band really depends on the extent of overexpression, suggesting that under our experimental conditions, EMRE overexpression is not high enough to overwhelm the mitochondrial protein processing machinery.

Figure 3A original Western blots

d) *What was the rationale for choosing the specific EMRE and MCU residues for cys-mutagenesis? If it was based on the hMCU/EMRE structure, this should be stated.*

Response: As we mentioned above, the majority of these crosslinking experiments were done before the structure paper came out. We apologize we did not make it clear how we decided to focus on these residues. We followed the following workflow to identify residues that crosslink, which can be briefly described as a partial cysteine-scanning strategy.

(a) After making a cysteine-free MCU construct, we selected one MCU amino acid residue at a time to be the sole cysteine residue. Several MCU residues on both transmembrane domains were experimentally tested to determine their ability to crosslink to EMRE. We confirmed that these cysteine mutations did not alter MCU protein function.

(b) We “scanned” several individual EMRE residues that had been mutated to cysteine for their ability to crosslink to the one MCU cysteine residue. Instead of testing all EMRE transmembrane residues for their ability to crosslink to the one MCU cysteine residue, we prioritized the EMRE residues that are likely to be in the same depth in the membrane as the selected MCU residue. We “guessed” their relative depth by the knowledge that a helical turn within the membrane is 3-4 amino acids long. We did not mutate EMRE amino acids 69 and 81 to cysteines as mutation of aa69 or aa81 to alanine caused loss of EMRE function and we predicted that mutating them to cysteine will have a similar effect.

e) *What amino acids replaced the cys in the cys-less MCU?*

Response: We apologize for omitting this sequence. It is now added to the manuscript. MCU has 2 cysteine residues (at amino acids 26 and 33) before the predicted MTS (ending at 56aa), these were mutated to glycine residues and did not affect protein localization.

It has 3 more cysteines at aa 66, 97 and 191.

C66 is an evolutionarily conserved residue. It was mutated to a Y. We decided not to mutate this residue to an S, as C66 is next to an ST sequence.

C97 is an evolutionarily conserved residue. It was mutated to a G. S92 residue nearby was shown to be phosphorylated. To avoid phosphorylation of this residue by a nearby kinase we chose to mutate C97 to a G.

C191 was mutated to an R due to presence of an R at this position in *Xenopus*. This indicated that an R at this position is likely to be tolerated.

f) In Fig 3C,D, please state what antibody was used for the IP blots.

Response: We now indicate that these are FLAG IPs in these figures. All other experimental details are also noted in the figure legends.

g) In Fig 4C. The authors make conclusions about "flexibility in this region of the complex". What do the authors mean? How does cross-linking to more than one residue provide information about protein dynamics or stability? There is a conclusion on page 13 that EDD is "flexible", but what is the evidence? And that it blocks exit from the pore...what is the evidence? There is too much speculation without experimental evidence.

Response: As the reviewer stated, cross-linking of one residue of MCU to more than one residue of EMRE does not provide information about protein dynamics or stability overall. Such crosslinking data can be explained by two possible scenarios (using MCU E296C data as example): (1) MCU E296C changes its relative localization/ orientation depending on conditions (2) MCU E296C has two static conformations on two different MCU peptides.

In this manuscript, we favored the first scenario, because CeMCU NMR structure and high-resolution structures of fungal MCU homologs (PMID: 29954988, 29995855, 29995857) showed that the region that coincides with EDD is disordered. We now add these fungal MCU structures with their "EDD" domains highlighted as a new supplementary figure (Supplementary Figure 7). And we change the main text to state that our flexibility claim comes from these fungal structures and our crosslinking data would be consistent with this. "To understand the significance of EMRE-EDD interaction for uniporter function, we identified the homologous EDD region in four fungal MCU homologs whose high-resolution structures have been published and highlighted EDD in these structures (**Supplementary Figure 7**). Surprisingly, EDD appeared flexible in these structures. Based on this observation and our crosslinking data, we posit that binding of EMRE stabilizes this otherwise flexible region and allows exit of Ca^{2+} ions from the pore".

Supplementary Figure 7: The region homologous to EDD appears disordered in fungi

High-resolution structures of MCU tetramers from four different fungal species are shown. The region that is homologous to EDD or amino acids that surround the EDD in these structures are shown in red. Dotted blue lines show amino acids that are omitted in these structures, all of which overlap with EDD. In the *M. acridum* MCU structure, one MCU chain shows alpha helical EDD, whereas the same region in the neighboring chain is not structured. PDB IDs of these structures are as follows: *M. acridum* (6C5W); *N. crassa* (5KUJ); *N. fischeri* (6D7W); *C. europaea* (6DNF)

h) P13. The authors suggest that they have demonstrated that the "N-terminus of EMRE is critical for the EMRE-EDD interaction", implying, based on the earlier part of the sentence, that it is critical for the EMRE-MCU interaction, but I don't think that this has been shown here. Furthermore, they state in the Discussion that there are likely additional extended interactions between MCU and EMRE

Response: We apologize for the ambiguous wording in this section. In experiments shown in Figures 4E, F and G, we "take away" the N terminus of HsEMRE by swapping it with that of CeEMRE, and this eliminates HsEMRE-HsMCU interaction, suggesting that this N terminus region of EMRE is required for EMRE-MCU interaction. When CeEDD is introduced to HsMCU to "put back" a binding partner for CeEMRE N terminus, this restores the lost interaction. These experiments suggest that N terminus of EMRE binds to EDD. This conclusion is corroborated by cysteine crosslinking between EMRE N terminus residues aa62 and aa59 with EDD. We re-wrote this section in the manuscript.

"Previous studies suggested that a small portion of N-terminal domain of EMRE is dispensable for MCU-EMRE interaction (38). However, in contrast, our crosslinking data show that the N-terminus of EMRE directly interacts with MCU. To determine the importance of this region for uniporter function and the stability of EMRE-MCU interaction, we generated chimeric proteins using HsMCU, HsEMRE, *C. elegans* MCU (CeMCU) and *C. elegans* EMRE (CeEMRE) as shown in **Figure 4E**."

13. Fig 4E-G are interesting. Can the authors explain the rationale for choosing *C. elegans*? What is the sequence conservation/divergence in these regions that might account for these results?

Response: We choose *C. elegans* as we knew that from our own unpublished data and from Tsai et al, Elife that *C. elegans* MCU (CeMCU) and CeEMRE will form a functional channel when expressed in MCU KO HEK 293T cells and their cDNAs were readily available. We also knew that (unpublished data) CeMCU does not bind to HsEMRE, making this system a good model to ask whether EMRE-EDD interaction is critical for EMRE-MCU interaction.

In the revised manuscript, we explained our rationale better by adding the following section We also provide sequence alignment of CeMCU- HsMCU and CeEMRE-HsMCU and highlight the sequences that were swapped (**Supplementary Figure 8**).

"Previous studies suggested that a small portion of N-terminal domain of EMRE is dispensable for MCU-EMRE interaction (38). However, in contrast, our crosslinking data show that the N-terminus of EMRE directly interacts with MCU. To determine the importance of this region for uniporter function and the stability of EMRE-MCU interaction, we generated chimeric proteins using HsMCU, HsEMRE, *C. elegans* MCU (CeMCU) and *C. elegans* EMRE (CeEMRE) as shown in **Figure 4E**. When expressed in MCU KO cells, CeMCU and CeEMRE form a functional channel, but HsMCU and CeEMRE are not compatible (38). Sequence alignment of MCU and EMRE from human and *C. elegans* are shown in **Supplementary Figure 8**. This system allowed us to test whether EDD and N terminus of EMRE contribute to MCU-EMRE interaction and channel function. HsEMRE with CeEMRE N terminal domain (HsEMRE^{CeNterm}) did not bind to HsMCU. When HsEMRE^{CeNterm} was expressed with HsMCU with CeEDD (HsMCU^{CeEDD}), the two proteins interacted and supported mitochondrial Ca²⁺ uptake (**Figures 4F and 4G**). We conclude that EDD and EMRE N-terminus interactions are necessary to form a stable association between MCU and EMRE and to form a functional pore."

Supplementary Figure 8: Alignment of human and C.elegans MCU TM1, TM2 and EDD, and EMRE N terminal domain.

Alignments of MCU or EMRE from human and C. elegans were done using CLUSTALW and amino acids were color coded using Boxshade. Black boxes show identical amino acids, gray boxes show similar amino acids. Purple box indicated EDD. Arrowheads indicate HsMCU and HsEMRE amino acids that crosslink.

14. I think that reference to the tetrameric JML in the Wang et al structure as "the Ca²⁺ exit site" of MCU is premature, even though Wang et al made the same claim. There is no evidence that Ca²⁺ needs to traverse this portal to enter the matrix, or that it even does. The structure is suggestive, but without functional evidence it is still an hypothesis. Perhaps the use of a word such as "putative" might be prudent. Furthermore, the authors state that chimera 9 has a constitutively open "Ca²⁺ gate distal to the pore", but it has not been shown that it is indeed a gate, nor that it is either constitutively closed or open under different conditions here. I think caution is warranted until there are experimental data.

Response: We agree with the reviewer that this site has not been shown to be a gate. We eliminated all references that refer to this site as a "gate".

15. The authors interpret lack of apparent inhibition by mutant MICU2 of MCU activity as evidence that MICU1/2 modulates the "gate distal to the pore", but lack of inhibition does not imply that MICU1/2 does not regulate Ca²⁺ access into the pore, i.e. there is no evidence to support this statement. Another way of thinking about the data is that the mutant chimera 9 channel modulates the channel structure in a way that impinges on MICU1/2 regulation. Additional experimentation would be necessary to dissect these alternatives.

Response: We agree with all the reviewers that the data shown in Figure 5 is preliminary and warrants more experiments. We removed Figure 5 from the manuscript, which now only focuses of mechanism of EMRE function in the complex.

16. P17, line 7, please insert a reference citation.

Response: This reference is now added.

17. Discussion paragraph 2. The human MCU/EMRE structure isn't mentioned at all here, whereas it's the focus of this study. Clearly, the Wang et al structure informed many of the experiments in this manuscript and the authors should acknowledge this.

Response: Again, we respectfully disagree with this assertion – the referee was clearly not aware of our bioRxiv submission which was dated five days after the submission of Wang. Now in the revised Discussion we discuss Wang et al structure more and compare/contrast our findings.

18. P17. The authors state that fungal MCU channels have lower conductance than animal MCU, but what is the evidence for this...it is not present in the references cited.

Response: Ernesto Carafoli and Albert Lehninger reported this as early as 1971 in a paper in which they also reported that yeast *S. cerevisiae* have no uniporter activity. In fact this paper was crucial for the Mootha lab's initial phylogenetics-inspired discovery of the uniporter machinery. In addition to the references cited, we added the papers listed below which show low or essentially no calcium uptake by fungal MCU homologs.

Gonçalves, A. P. et al. Involvement of mitochondrial proteins in calcium signaling and cell death induced by staurosporine in *Neurospora crassa*. *Biochim. Biophys. Acta - Bioenerg.* 1847, 1064–1074 (2015).

Wettmarshausen, J. et al. MICU1 Confers Protection from MCU-Dependent Manganese Toxicity. *Cell Rep.* 25, 1425-1435.e7 (2018).

Discovery of EMRE in fungi resolves the true evolutionary history of the mitochondrial calcium uniporter
Alexandros A. Pittis^{1,2,5,7#}, Valerie Goh^{3#}, Alberto Cebrian-Serrano³, Jennifer Wettmarshausen³, Fabiana Perocchi^{3,4*} & Toni Gabaldón^{1,2,6,8*} <https://www.biorxiv.org/content/10.1101/2020.03.24.006015v1.full.pdf>

Referee #2:

This manuscript reports analysis to identify the EMRE and MCU interacting domains based on the use of chimeras between human and dictyostelium MCU and partial cysteine scan. They also use one of the chimera to examine a new potential role of MICU1/2 in the regulation of the MCU pore.

Although the result are fairly clean, unfortunately, they add very little to the structure of MCU/EMRE published in May of last year. The structure provides a more detailed information and makes it quite difficult to support publication of this manuscript in this journal.

Response: We thank the reviewer for critical reading of our manuscript. Although publication of the MCU/EMRE structure did hurt the novelty of our manuscript, we would like to stress that we deposited an earlier version of this manuscript on Biorxiv preprint server five days after MCU/EMRE structure was published, and our functional data for the most part agrees with the structure paper and complements it. Our work includes extensive experiments, performed without the bias of a structure, and hence, provides independent experimental data that can be viewed as a “companion” for these structural studies. The majority of our experimental findings are consistent with the Wang et al structure. Many of our experimental findings help to inform the structure, and conversely, we have some findings – generated again in an unbiased way – that do not agree with the structure and hence could be revealing some additional novelty.

Novel findings in our paper is summarized here:

a) We identify EDD, which is overlapping but distinct from JML. Comparison of functional data between different chimeras (compare chimera 9 to chimeras 6 and 7) clearly show the importance of residues outside of JML for EMRE function.

(b) We show more extensive interactions between EMRE and MCU pore forming TM2 that were not detected in the structure. Wang et al claims that EMRE and MCU interact only at TM1 and around the JML in TM2 (around MCU amino acids 280- 300). We find robust and reproducible interaction between MCU and EMRE at amino acid 270, which is closer to the intermembrane space. We now discuss in the manuscript that these differences can be due to technical issues or dynamic nature of the complex *in vivo*. These results also caution the community against taking the structure as the only and ultimate confirmation for MCU-EMRE complex; argues for the presence of other, functionally important MCU-EMRE interactions *in vivo*; and makes a strong case for additional functional experiments to understand how this interesting calcium channel functions at the molecular level *in vivo*.

(c) We provide data for the first time that interactions between EMRE N-terminal domain and MCU EDD are important for both binding and function of the channel (Figures 4F and 4G). Previous studies suggested that MCU TM1 and TM2 are the determinants, but our data clearly show that even though TM domains of MCU and EMRE are wild type sequences from the same species, incompatibility at EMRE’s N-terminal domain and MCU’s EDD lead to loss of binding between the two.

The only additional and new findings are in Figure 5, however, they are preliminary as presented. The lack of effect of MICU2 can be for multiple reason such as altered interaction of the chimera with MICU2 (lower affinity), altered interaction with MICU1 that is required for the effect of MICU2, altered Ca²⁺-dependence of Ca²⁺ uptake by the chimera, among others. More rigorous characterization of the chimera, including biophysical analysis of chimera pore, is needed to claim a role for MICUs on channel gating through interaction between MCU-EMRE.

Response: We acknowledge the preliminary nature of the experiments in Figure 5 and have decided to remove this figure from the current manuscript.

Referee #3:

Mitochondrial calcium uptake is critical for mitochondrial and cellular function. The uptake route relies on the channel MCU which interacts with numerous regulatory/accessory proteins. One of these is the protein EMRE which interestingly appears not to be co-conserved with MCU. Still EMRE is critical for MCU function in organisms containing EMRE. This implies that there might be features in (human) MCU that are important for its functional dependence on EMRE. In this study, the authors set out to identify these regions using chimeras of EMRE-dependent and EMRE-independent MCU variants. They thereby identify a region in MCU critical for EMRE interaction and characterize it further biochemically.

This is a very nice piece of high-quality biochemical work. My major concern affects the conceptual novelty of the findings: the interaction site of EMRE and MCU has already been previously identified in MCU (Wang et al) - although in a slightly different amino acid window. The current study expands on this but does not significantly address the in my opinion critical and really novel question of the evolutionary and regulatory significance of EMRE.

Response: We thank the reviewer for their constructive criticism. We agree with the reviewer that evolutionary and regulatory significance of EMRE is a critical question in the field and Wang et al. hurt the novelty of this manuscript. Please note that the pre-print version of our current paper was published on the bioRxiv only five days after the publication of the Wang et al structure paper in 2019. As this reviewer acknowledges, our current paper comprises “*very nice piece of high-quality biochemical work*” and as it was generated without knowledge of the structure, can be viewed as an experimental companion to the structures that are now being reported with minimal experimental validation. We would like to point out that 6/8 of the crosslinking sites that we are reporting are consistent with the Wang et al, however, 2/8 are not readily explained. While they could represent artifacts of cross-linking (as we acknowledge), they are robust as the other four and could represent latent conformational changes not observed in cryo-EM. As EMRE expression can change independent of MCU, including in some disease states, we end by speculating that EMRE evolved to afford an extra layer of regulation.

Further points:

Fig. 2: complementation studies with MCU chimeras were performed in either MCU or EMRE single KO cells. Why did the authors not use MCU-EMRE double knockout cells to exclude e.g. interference of endogenous MCU with chimeras in EMRE KO cells?

Response: We agree with the reviewer that some of the experiments shown here would have been easier to perform in MCU/EMRE double knockout cells, however, we did not have access to these cells when we started the chimera experiments (dating back to 2015). This said, we would have expressed the chimeras in single KO cells either way, since expressing each chimera in MCU KO cells was done to test whether particular chimera was functional or not. If used double KO cells and observed no calcium uptake, we should not have known whether this was due to chimera being non-functional or due to absence of EMRE and would have expressed them in single KO cells anyways. So, in this particular case, we don't think using double KO cells would bring an advantage.

Fig. 4: Cysteine crosslinking data (interaction between far-away residues) - the authors interpreted their findings as "flexible nature" of the interaction. How can they exclude artefacts from the experiment? What is the functional relevance of this flexible interaction? Is the interaction dynamic?

Response: We thank the reviewer for this very insightful question and apologize that we did not explain the significance of these findings better. We now acknowledge in the revised manuscript that the robust crosslinking we observe could be experimental artifacts or could be due to different conformational states or *in vivo* dynamics of the uniporter that are hard to capture in a high-resolution structure. We believe this is an important point in channel regulation which is overlooked in structural studies. Addressing the functional relevance of this flexible interaction requires further experiments that are beyond the scope of this manuscript.

Figure 5: MICU1 binding to MCU does not result in closing of chimera 9. This implies that EMRE is hierarchically more important for the control of opening and closing. However, compared to MICU1, its binding to MCU appears not to be dynamically regulated? Can the authors further elucidate on this and hypothesize how the non-dynamic binding of EMRE helps in regulating MCU activity.

Response: We agree with the reviewer that regulation of MCU by EMRE and MICU1 is one of the main questions that remain controversial in the field. Our preliminary functional experiments point to a model where MICU1 binding allosterically regulates EMRE-MCU interaction at the channel exit. However, given that several reviewers commented on the preliminary nature of these experiments, and the COVID-19 pandemic, we decided to remove our figure that addresses channel regulation by MICU1 from this manuscript and focus on this important question in a separate manuscript.

p.3, 2nd paragraph: "Intermembrane" should read "intermembrane space"

Response: We thank the reviewer for catching this. We corrected this mistake.

July 28, 2020

RE: Life Science Alliance Manuscript #LSA-2020-00718-TR

Dr. Yasemin Sancak
University of Washington
Department of Pharmacology
1959 NE Pacific Street
Rm K536B
Seattle, Washington 98195-7750

Dear Dr. Sancak,

Thank you for submitting your revised manuscript entitled "Evolutionary divergence reveals the molecular basis of EMRE dependence of the human MCU". We would be happy to publish your paper in Life Science Alliance pending final revisions necessary to meet our formatting guidelines.

-please have the secondary corresponding author add their ORCID ID--they should have received instructions on how to do so

A. FINAL FILES:

B. MANUSCRIPT ORGANIZATION AND FORMATTING:

Sincerely,

Reilly Lorenz
Editorial Office Life Science Alliance
Meyerhofstr. 1
69117 Heidelberg, Germany
t +49 6221 8891 414
e contact@life-science-alliance.org
www.life-science-alliance.org

July 29, 2020

RE: Life Science Alliance Manuscript #LSA-2020-00718-TRR

Dr. Yasemin Sancak
University of Washington
Department of Pharmacology
1959 NE Pacific Street
Rm K536B
Seattle, Washington 98195-7750

Dear Dr. Sancak,

Thank you for submitting your Research Article entitled "Evolutionary divergence reveals the molecular basis of EMRE dependence of the human MCU". It is a pleasure to let you know that your manuscript is now accepted for publication in Life Science Alliance. Congratulations on this interesting work.

DISTRIBUTION OF MATERIALS:

Again, congratulations on a very nice paper. I hope you found the review process to be constructive and are pleased with how the manuscript was handled editorially. We look forward to future exciting submissions from your lab.

Sincerely,

Reilly Lorenz
Editorial Office Life Science Alliance
Meyerhofstr. 1
69117 Heidelberg, Germany
t +49 6221 8891 414
e contact@life-science-alliance.org
www.life-science-alliance.org